# Intrinsic and extrinsic cues time somite progenitor contribution to the vertebrate primary body axis

Lara Busby[1,2], Guillermo Serrano Nájera[1], Benjamin John Steventon[1]*

[1]Department of Genetics, University of Cambridge, Cambridge, United Kingdom; [2]Department of Molecular and Cell Biology, University of California, Berkeley, Berkeley, United States

**Abstract** During embryonic development, the timing of events at the cellular level must be coordinated across multiple length scales to ensure the formation of a well-proportioned body plan. This is clear during somitogenesis, where progenitors must be allocated to the axis over time whilst maintaining a progenitor population for continued elaboration of the body plan. However, the relative importance of intrinsic and extrinsic signals in timing progenitor addition at the single-cell level is not yet understood. Heterochronic grafts from older to younger embryos have suggested a level of intrinsic timing whereby later staged cells contribute to more posterior portions of the axis. To determine the precise step at which cells are delayed, we performed single-cell transcriptomic analysis on heterochronic grafts of somite progenitors in the chicken embryo. This revealed a previously undescribed cell state within which heterochronic grafted cells are stalled. The delayed exit of older cells from this state correlates with expression of posterior *Hox* genes. Using grafting and explant culture, we find that both *Hox* gene expression and the migratory capabilities of progenitor populations are intrinsically regulated at the population level. However, by grafting varied sizes of tissue, we find that small heterochronic grafts disperse more readily and contribute to more anterior portions of the body axis while still maintaining *Hox* gene expression. This enhanced dispersion is not replicated in explant culture, suggesting that it is a consequence of interaction between host and donor tissue and thus extrinsic to the donor tissue. Therefore, we demonstrate that the timing of cell dispersion and resulting axis contribution is impacted by a combination of both intrinsic and extrinsic cues.

*For correspondence:
bjs57@cam.ac.uk

Competing interest: The authors declare that no competing interests exist.

## Editor's evaluation

This interesting study provides valuable data exploring how progenitors control their contribution to somitogenesis. By combining classic embryology techniques with single-cell sequencing, the authors describe novel cell states that might help understand progenitor population dynamics. This manuscript will be of interest to researchers in the development field who want to better understand hox control and influence during axial elongation.

## Introduction

Timing is a fundamental concept in developmental biology: development is inherently dynamic and occurs with reproducible timing for the embryos of a given species (*Busby and Steventon, 2021*; *Duboule, 2022*; *Ebisuya and Briscoe, 2018*; *Raff, 2006*; *Rayon, 2023*). Developmental tempo refers to the cell-intrinsic features that produce species-specific developmental speeds, including rates of transcription, translation, and protein turnover (*Matsuda et al., 2020*; *Rayon et al., 2020*). In the

context of an embryo, cells which have intrinsic features must regulate their dynamics in accordance with their surroundings to produce emergent properties in a manner that is not well understood. Therefore, the relative balance of intrinsic and extrinsic cues in timing events in development is an important question that has been investigated in contexts including limb development and the vertebrate tailbud (*Chinnaiya et al., 2014*; *Fulton et al., 2022*; *Pickering et al., 2018*; *Saiz-Lopez et al., 2015*; *Saiz-Lopez et al., 2017*; *Perez et al., 2023*; *Stainton and Towers, 2022*).

The vertebrate primary body axis comprises a set of highly conserved axial and paraxial components, namely the midline notochord, dorsal neural tube, and pairs of somites on either side of the midline. These structures are produced in a defined anterior to posterior sequence by the controlled allocation of axial progenitor cells to the body axis over time. In the avian embryo, this occurs through the passage of cells through the blastopore-analogous structures of the primitive streak (PS) and node, which will go on to form a structure termed the tailbud, found at the caudal extremity of the embryo (*Brown and Storey, 2000*; *Catala et al., 1996*; *Gray and Dale, 2010*; *Iimura et al., 2007*; *McGrew et al., 2008*; *Schoenwolf, 1977*; *Selleck and Stern, 1991*; *Yang et al., 2002*). Thus, a population of progenitor cells is maintained within the embryo throughout the period of axis elongation. Crucially, the contribution of cells from progenitor populations to the axis occurs in a gradual manner, with cells streaming through the PS or later from the tailbud over a period of several days (*Hambuger and Hamilton, 1951*). The mechanisms which underlie this gradual streaming behaviour are unknown, though a role for *Hox* genes in timing axis contribution of progenitor cells has been proposed (*Deschamps and Duboule, 2017*; *Iimura and Pourquié, 2006*). During development of the vertebrate posterior body, axial progenitor cells express *Hox* genes in a differential manner over time – a phenomenon termed temporal collinearity ('the Hox Clock') (*Deschamps and Duboule, 2017*; *Izpisúa-Belmonte et al., 1991*). Specifically, as development progresses, axial progenitor populations express progressively more 5′ *Hox* gene complements. Previous work has implicated *Hox* genes in the control of cell behaviour during posterior body development: the overexpression of various 5′ *Hox* genes in axial progenitor cells of the avian PS is sufficient to delay their contribution to the body axis (*Iimura and Pourquié, 2006*). More 5′ *Hox* genes delay the timing of cell contribution to a greater extent than more 3′ *Hox* genes (*Iimura and Pourquié, 2006*).

Importantly, the relative role of cell-intrinsic and -extrinsic cues in timing the decision of individual cells to enter the body axis is unknown. The delayed contribution of heterochronic grafts of tissue from older to younger embryos relative to stage-matched grafts (*Iimura and Pourquié, 2006*; *Tam and Tan, 1992*) suggests a degree of intrinsic control over axis contribution timing, but the extent to which this is true remains to be understood. In this article, we exploit the rich toolkit provided by experimental embryology (*Busby et al., 2022*) in conjunction with next-generation sequencing methods and multiplexed staining to investigate the extent to which axial progenitor gene expression and behaviour are intrinsically or extrinsically regulated.

A progenitor population that produces the medial portion of somites has been described in the avian embryo, residing at the 90% region of the PS (*Iimura et al., 2007*; *Psychoyos and Stern, 1996*). We chose to focus on this cell population, herein termed the medial somite progenitor (MSP) population, as a model for understanding the timing of cell contribution to the body axis. In particular, we focus our analyses on HH4 (definitive PS stage) and HH8 (4 somite-stage) embryo for several reasons. Whilst the somite progenitor population is continuous across these stages, it is known that the first 4–5 somites of the avian axis are distinct from more posterior ones – they are formed simultaneously (*Dias et al., 2014*), lack the rostro-caudal sub-patterning present in more posterior somites (*Rodrigues et al., 2006*), are not invaded by neural crest cells which give rise to ganglia (*Lim et al., 1987*), and do not give rise to segmented structures but instead to the occipital and sphenoid bones of the skull (*Couly et al., 1993*; *Huang et al., 2000*). Cells of the HH4 MSP region give rise to the occipital somites (*Psychoyos and Stern, 1996*), whereas at HH8 these somites have formed already (*Hambuger and Hamilton, 1951*), and so the progenitor region exclusively produces trunk and tail somites. It is therefore interesting to consider possible differences between the dynamics and behaviour of the progenitor region between HH8 and HH4, and whether these are maintained irrespective of the environment that the tissue is placed in. In normal development, the context in which the HH4 and HH8 MSP regions allocate cells to the axis is very different – at HH4 (primary gastrulation) the mesendoderm represents an unbounded compartment that cells move into after ingression through the PS, whereas at HH8 a clear pre-somitic

mesoderm is formed and cells are allocated to this bounded compartment. Despite the substantial differences between occipital and trunk somite formation, it remains unknown whether these modes of somitogenesis result from changing environmental inputs or changes to the progenitor population itself over time.

We find that heterochronic grafts of somitic progenitors from HH8 to HH4 are delayed relative to stage-matched grafts at a previously undescribed step that we term 'dispersion', occurring after cells pass through the PS and enabling substantial mixing between donor and host tissue in the mesendodermal compartment. By culturing MSP tissue ex vivo on fibronectin, we find that the delayed migration is an intrinsic property of HH8 tissue and can account for the delay in axis contribution of heterochronic grafts. Surveying *Hox* gene expression in the MSP region at HH4 and HH8 reveals key differences in gene expression, which are maintained in grafted HH8 tissue in the HH4 environment, suggesting that *Hox* gene expression is also an intrinsic property of the tissue. Together, this work demonstrates that a balance of tissue-intrinsic and -extrinsic timers controls *Hox* gene expression and cell dispersion during progenitor addition to the body axis.

## Results
### Delayed contribution of heterochronic grafts is characterised by a lack of dispersion of cells from the primitive streak

To ask whether the behaviour of progenitor cells differs before and after the onset of node regression (cessation of primary gastrulation), we compared homochronic HH4 to HH4 grafts with HH8 to HH4 grafts. Homochronic grafts of MSP cells were performed at HH4 (*Hambuger and Hamilton, 1951*) from a GFP transgenic line (*McGrew et al., 2008*) to a wild-type embryo (*Figure 1A*). Over time, the gradual contribution of cells deriving from the MSP population to the body axis can be observed (*Figure 1B–G*). At each timepoint, a population of MSP cells remains closely associated with the morphological remnant of the node during node regression after HH5 (indicated by the yellow boxes in *Figure 1B–G*), whilst other cells ingress through the streak and begin to migrate towards their final location in the medial portion of a somite (*Figure 1G' and G'''*). At the final timepoint imaged (24 hr after grafting), GFP-expressing donor cells can still be found in the progenitor region, by which time the embryo has reached tailbud stage (*Figure 1G''*).

We then performed grafts of the MSP region from HH8 embryos into the HH4 embryo (*Figure 1L*, extended data in *Figure 1—figure supplement 1*). Within 3 hr, GFP donor cells in homochronic grafts have spread appreciably from the graft site (*Figure 1I vs. J*), whereas the location of GFP donor cells in heterochronic grafts does not differ substantially from immediately post-grafting (compare *Figure 1M* with *Figure 1N*). The change in area occupied by the graft (between 0 and 3 hr post-grafting) was quantified and found to be statistically significant between the two graft types (*Figure 1P*). Importantly, this delay in dispersion correlates with a difference in anteroposterior somite contribution: whilst homochronic grafts invariably have their most anterior axis contribution in somite 1, heterochronic grafts typically have their most anterior contribution in somite 4 or 5 (*Figure 1K vs. O*, quantification in *Figure 1Q*).

To determine whether the lack of cell dispersion in heterochronic grafts is reminiscent of HH8 cell behaviour in the HH8 environment, we performed stage-matched HH8 to HH8 grafts and allowed them to grow for 24 hr (*Figure 1R–S*). Importantly, stage-matched HH8 to HH8 grafts mix with surrounding host tissue (*Figure 1S'*), suggesting that the limited dispersion of heterochronic grafts is associated with the combination of HH8 tissue with the HH4 environment. At 24 hr after grafting, HH8 to HH8 stage-matched grafts are primarily found in the presomitic mesoderm (PSM), meaning that they contribute to somites posterior to somite 10 (*Figure 1S*). Therefore, while heterochronic grafts (HH8 to HH4) are delayed relative to HH4-to-HH4 grafts in their axis contribution, these cells nonetheless contribute to a more anterior portion of the axis than they would in their normal developmental context.

These results show that HH8 progenitor cells exhibit substantially different behaviour in the HH4 environment relative to stage-matched grafts as early as 3 hr after grafting. We therefore focused on the initial ingression and spreading of cells from the PS to elucidate which cell state transitions are key for the timing of somite progenitor addition.

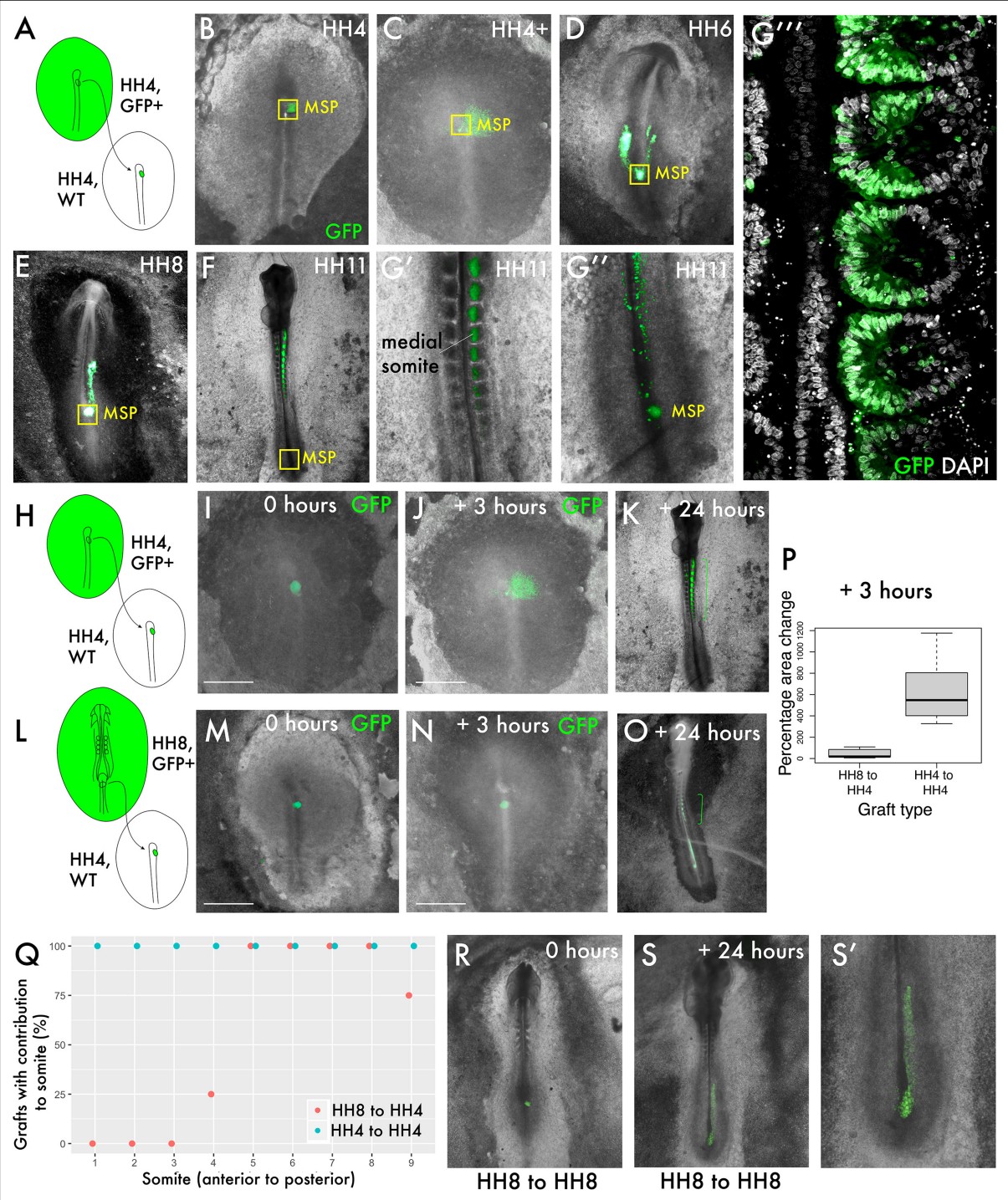

**Figure 1.** Delayed contribution of heterochronic grafts is characterised by a lack of dispersion of cells from the primitive streak (PS). (**A**) Homochronic grafts of the 90% PS region were performed from GFP-expressing HH4 embryos to wild-type HH4 embryos. (**B–G"**) Combined GFP/brightfield images of grafts at various times between 0 and 24 hr after grafting reveal the behaviours of this population during body axis development. As the node regresses posteriorly after HH5, a subpopulation of the medial somite progenitor (MSP) region remains associated closely with the morphological remnant of the node (yellow box 'MSP'), whilst other cells clearly part from this population and move towards the anterior of the embryo, where the somites will form. At 24 hr post-graft, embryos are approximately HH10-11 and GFP cells are clearly located in the medial portion of the somites (**G'**), close to the midline. At this timepoint, a population of GFP+ cells remain at the posterior progenitor region of the embryo (the tailbud) (**G"**). (**G'''**) is a single-slice confocal image of the graft in (**G', G"**) stained with DAPI showing the precise localisation of grafted cells (GFP) in the medial portion of the somite. (**H**) and (**L**) are schematic representations of the two graft types, showing homochronic HH4-HH4 and heterochronic HH8-HH4 grafts, respectively. (**I**) and (**M**) show overlaid GFP/brightfield images of homochronic (HH4-HH4) and heterochronic (HH8-HH4) grafts at 0 hr after grafting.

*Figure 1 continued on next page*

Figure 1 continued

(**J**) and (**N**) show the same grafts at 3 hr, when the homochronic graft has spread substantially from the initial graft site but the heterochronic graft appears very similar to the 0 hr image. (**K**) and (**O**) are combined GFP/brightfield images of homochronic and heterochronic grafts, respectively, at 24 hr after grafting. The green brackets in (**O**) and (**P**) indicate the somite contribution of the two grafts. (**P**) Quantification of the change in area of GFP-positive grafted tissue in homochronic and heterochronic grafts at 3 hr vs. 0 hr post-graft, represented as a boxplot. N = 6 for homochronic grafts and n = 5 for heterochronic grafts. (**Q**) is a scatterplot showing the somite contribution of HH4-HH4 and HH8-HH4 grafts, with the percentage of grafts that have cells in each somite plotted against somite number. The different colours of datapoint represent the type of graft (homochronic or heterochronic). N = 8 for homochronic grafts and n = 4 for heterochronic grafts. (**R–S**) Combined GFP/brightfield images of HH8-to-HH8 grafts at 0 and 24 hr after grafting, respectively. Scale bars in (**I**), (**J**), (**M**), and (**N**) represent 1 mm.

The online version of this article includes the following figure supplement(s) for figure 1:

**Figure supplement 1.** HH8 to HH4 graft time series.

## Single-cell analysis reveals a previously undescribed cell state within which heterochronic grafted cells are stalled

To investigate the cell states occupied by homochronic and heterochronic MSP grafts, we turned to single-cell RNA-sequencing (*Figure 2*). Grafted tissue was dissected, pooled, and dissociated, with each sample comprising a mixture of GFP+ donor cells and GFP- host cells (*Figure 2A*). Cells from all three samples were integrated to a single dataset and clustered, yielding 14 distinct clusters (*Figure 2B*). Published expression data (GEISHA – *Bell et al., 2004*; *Darnell et al., 2007*) was used to annotate these cluster identities (*Figure 2—figure supplement 1*, *Figure 2B and C*), and the expected cell types were present, including epiblast, ingressed mesoderm, endoderm, and Hensen's node. Note that there were four clusters in the dataset – 6, 8, 12, and 13 – that are found at the centre of the UMAP plot (*Figure 2B*). Differential gene expression analysis revealed that while they lacked the expected markers of ingressed mesoderm (as seen for clusters 2 and 5), they do express some markers of PS and early mesoderm (e.g. *Lhx1* and *Tnn2* [cluster 6]; *Lfng*, *Prtg*, *Cdh2*, and *Acvr2b* [cluster 12]; *Fabp5* and *Rac1* [cluster 8]; *Supplementary file 1a*), suggesting that they represent a transitory state. Splitting the dimensionality plot by sample (*Figure 2D–F*) shows some differences in the clusters populated in each sample – most notably, the HH4-4 sample lacks clusters 0, 4, and 7 which represent HH8 tissue (expressing posterior marker genes, including *Hoxb8* and *Cdx4*, *Figure 2C*; *Barak et al., 2012*; *Bell et al., 2004*; *Darnell et al., 2007*; *Joshi et al., 2019*). As the dataset comprises both host and donor cells in each instance, we plotted the expression of *eGFP* in each dataset to ask what the distribution of donor tissue is (*eGFP*+, *Figure 2G–I*) compared to host cells (eGFP-, *Figure 2G'–I'*). In HH4-HH4 control grafts, *eGFP*-expressing cells are found across the clusters present in the dataset mixed with *eGFP*-negative wild-type cells, consistent with host and donor tissue being equivalent in this instance (*Figure 2G and G'*). Note that a reduced sample was also prepared of HH4-HH4 3 hr grafts in which clustering yields intermixed eGFP-positive and -negative cells, suggesting that grafting does not cause substantial changes to the transcriptome (*Figure 2—figure supplement 2*). In HH8-4 grafts at 0 hr, *eGFP*-expressing cells are primarily concentrated in the left-hand clusters on the UMAP plot, in clusters 0, 4, and 7 (*Figure 2H*). This shows that HH8 tissue occupies a distinct region in gene expression space initially upon grafting. Strikingly, after 3 hr, HH8 *eGFP*-positive cells are enriched in the central clusters 6, 8, and 12 (*Figure 2I*). Importantly, surveying the HH4-HH4 homochronic graft dataset shows that this cell state is not unique to the heterochronic graft condition, though there is not a particular enrichment of cells in this state in normal development (*Figure 2D and G*). Thus, our scRNA-seq experiment reveals an undescribed cell state in mesodermal progenitor contribution to the body axis which HH8 cells accumulate in upon grafting.

## The transition from ingressed mesoderm to individual dispersed cells is a key step in gastrulation, associated with changes in cell adhesion

To ask what the cell states encompassed by clusters 6, 8, and 12 represent, we revisited our grafts to determine whether heterochronically grafted cells had undergone an epithelial-to-mesenchymal transition (EMT) and ingressed through the PS or whether they remained in the epiblast. Re-slicing Z-stacks of these grafts allowed us to survey the localisation of eGFP-positive cells in the dorsoventral axis (*Figure 3A and B*). In both homochronic and heterochronic grafts, eGFP-positive cells clearly reside in the lower, ventral mesendodermal cell layer rather than the more dorsal epiblast (*Figure 3A*

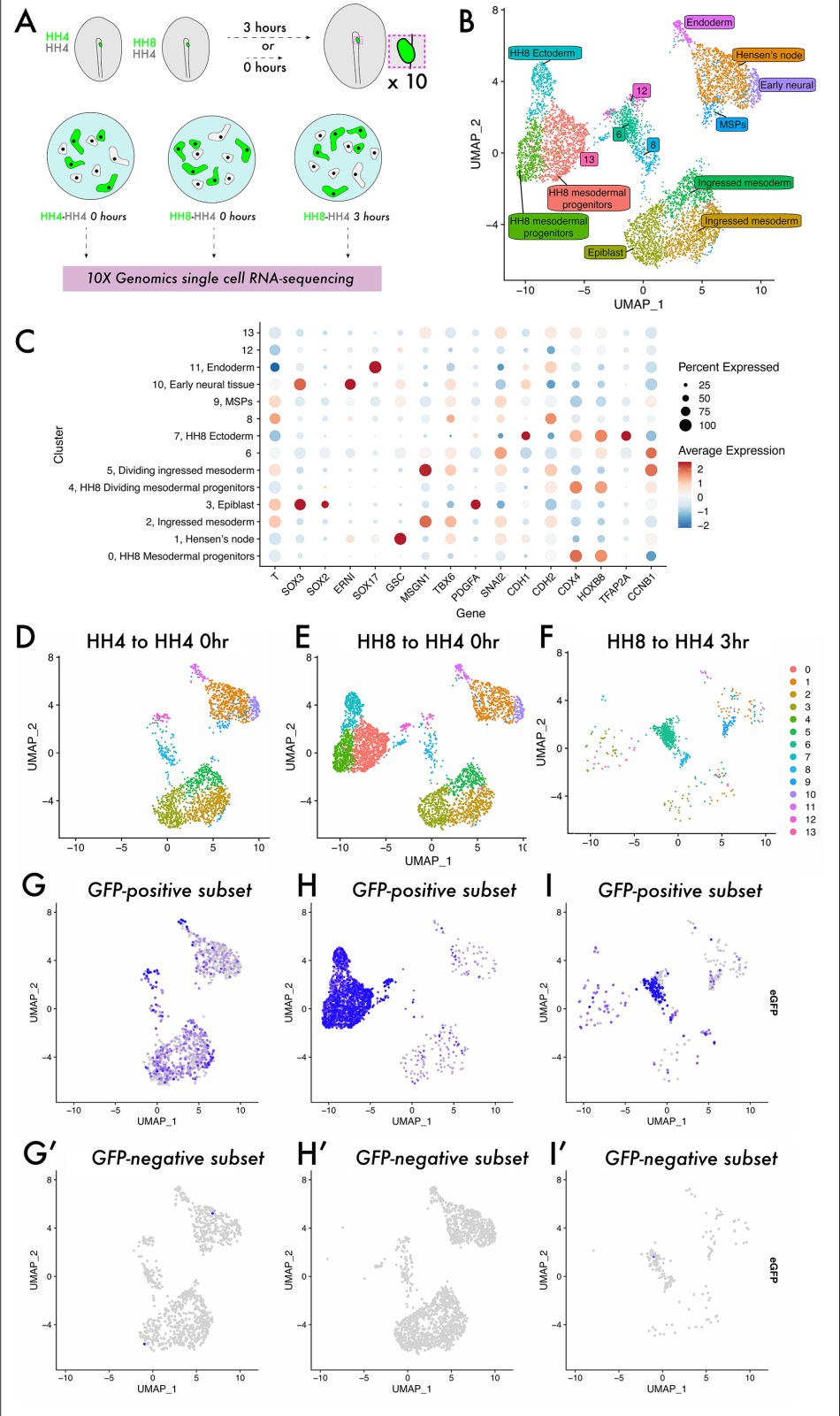

**Figure 2.** Single-cell analysis reveals a previously undescribed cell state within which heterochronic grafted cells are stalled. (**A**) Schematic representing single-cell RNA-sequencing experimental design. Grafts were performed from a GFP-expressing donor embryo to a wild-type host embryo and the region containing fluorescent tissue was dissected either 0 or 3 hr after grafting. Ten grafts were pooled for each condition and subjected to dissociation

*Figure 2 continued on next page*

*Figure 2 continued*

and single-cell RNA-sequencing. (**B**) UMAP plot showing cells from all three datasets clustered according to gene expression differences. The colour of each point on the plot represents the cluster to which it has been assigned, and labels denote cluster identities assigned according to the expression of previously described 'marker' genes. Note that clusters 6, 8, 12, and 13 were unable to be assigned a specific cell state identity based on described gene expression. (**C**) Dotplot showing the expression of genes in each cluster of the combined dataset. The appearance of each dot is a composite of the percentage of cells in that cluster with positive expression of the gene (size of dot) and the average quantitative level of gene expression (colour of dot) – see key on the far right of the plot. (**D–F**) UMAP plots of each sample, with the colour of each spot (cell) representing the cluster to which it has been assigned. (**G–I**) UMAP plots of each sample, with the colour of each spot (cell) representing the expression level of eGFP; more purple spots represent higher expression values. Cells with expression values >0.001 can be identified as donor cells deriving from the GFP-expressing transgenic embryo and are shown in (**G–I**). Cells with expression values <0.001 are identified as host cells and are shown in (**G'–I'**).

The online version of this article includes the following figure supplement(s) for figure 2:

**Figure supplement 1.** scRNA-seq annotation supporting data.

**Figure supplement 2.** HH4 to HH4 3 hr sample.

*and B*). When we surveyed the expression of *Snail2 (Snai2)* – considered a master regulator of EMT and required for mesodermal cells to ingress through the PS (*Nieto et al., 1994*) – we found its expression in cells of clusters 6, 8, and 12 (*Figure 3D*, compare with *Figure 3C*). Indeed, we confirmed the expression of the EMT-associated transcripts *Snai2* and *N-Cadherin (Cdh2)* (*Dady et al., 2012*) in grafted tissue by multiplexed hybridisation chain reaction (HCR) (*Figure 3E–F*) and found their expression in both homochronic and heterochronic grafts at 3 hr. To further probe the morphology of grafted cell populations, we fixed both homochronic and heterochronic grafts after 20–30 min, as soon as the tissue had integrated into the host embryo. These embryos were stained with phalloidin and immunostained for the basement membrane component laminin (*Figure 3G–H*). Homochronic grafts show a similar nuclear density to surrounding host tissue (*Figure 3G*). In addition, the basement membrane within the grafted tissue is incompletely broken down at 30 min (*Figure 3G"*) consistent with the grafting of full-thickness streak tissue comprising both epiblast and mesendodermal cells. In contrast, heterochronic grafts at 30 min have an elevated nuclear density compared to surrounding host tissue (*Figure 3H*) and are delineated by a circumferential ring of F-actin (*Figure 3H'''*). These results suggest that HH8 cells can undergo gene expression changes associated with EMT but are delayed in their migration away from the PS, in a step we will refer to as *cell dispersion*. The observed differences in laminin distribution and cell density immediately after grafting suggest that this could be a consequence of differences in tissue organisation of donor cell populations.

To further investigate what characterises the cell state that HH8 cells are stalled in, we performed Gene Ontology (GO) analysis on the list of genes differentially expressed in cluster 6 that are present across samples. Cluster 6 is where the majority of HH8 cells are found at 3 hr after grafting. We found a clear enrichment of genes associated with cell adhesion in this cluster (*Figure 3I*, full gene lists in *Supplementary file 1b*), suggesting that the delayed dispersion and axis contribution of grafted cells may be accounted for by differences in cell adhesion. To ask if these genes are differentially expressed between HH4- and HH8-derived cells (and thus could account for the different behaviour of these two populations upon grafting), we subsetted the scRNA-seq dataset to clusters 6, 8, and 12 only, annotated cells as either HH8- or HH4-derived (based on *eGFP* expression), and looked for genes differentially expressed between the two groups (schematic in *Figure 3J*). Note that there are subtle differences between the cell cycle state of HH8 and HH4 populations (*Figure 3—figure supplement 1B and D–E*) and so we regressed cell cycle-associated genes from the analysis to focus on other differences. The most significantly enriched GO term in this analysis is 'anteroposterior pattern specification' (*Figure 3K*, *Supplementary file 1c*), which annotates genes, including the *Hox* genes *Hoxa7* and *Hoxb8*. These genes exhibit substantial expression in HH8-derived cells but no expression in HH4-derived cells (*Figure 3L*). Other GO terms enriched in this analysis included epigenetic-related terms such as 'chromatin organization', 'DNA methylation' and 'C-5 methylation of cytosine' (*Figure 3K*, *Supplementary file 1c*), consistent with the described importance of chromatin remodelling in *Hox* gene activation in development (*Noordermeer et al., 2011*; *Soshnikova and Duboule, 2009*).

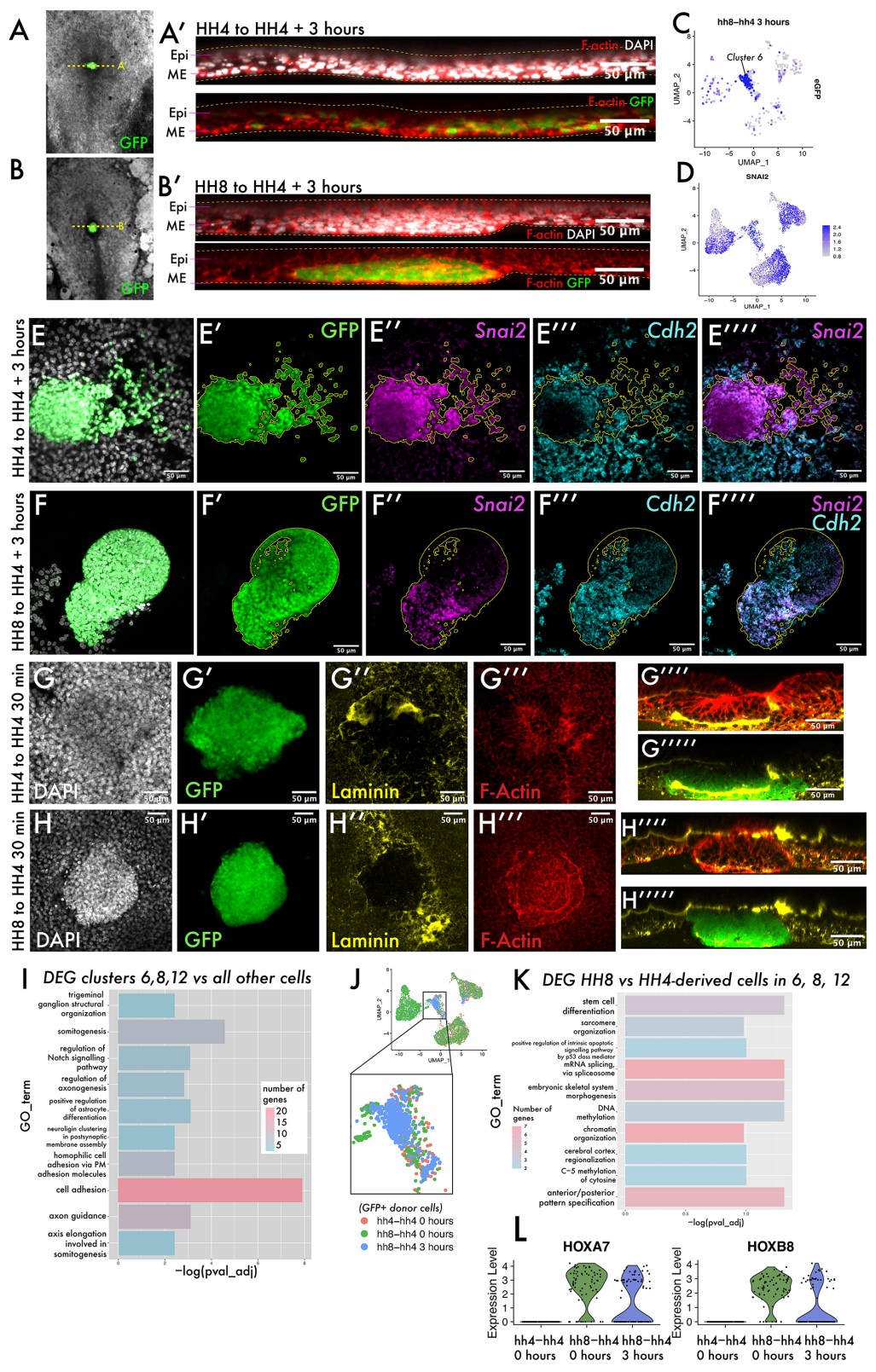

**Figure 3.** The transition from ingressed mesoderm to individual dispersed cells is an important step in gastrulation, associated with changes in cell adhesion. (**A–B**) Composite GFP/brightfield images of HH4-HH4 and HH8-HH4 grafts at 3 hr. Dotted yellow lines represent the plane of section shown in (**A'**) and (**B'**), respectively. (**A'**) and (**B'**) are summed slice projections (~10 μm) re-sliced Z-stacks of the grafts in (**A**) and (**B**), respectively, fixed

*Figure 3 continued on next page*

*Figure 3 continued*

and stained with phalloidin (red) and DAPI (white). Images in (**A'**) and (**B'**) are oriented with dorsal at the top and ventral at the bottom. (**C**) UMAP plot for HH8-HH4 sample showing eGFP expression. (**D**) UMAP plot of the full combined dataset showing Snail2 (Snai2) expression. (**E**) and (**F**) Summed slice Z-projections of HH4-HH4 and HH8-HH4 grafts stained by multiplexed hybridisation chain reaction (HCR) for the EMT markers Snai2 and N-cadherin (Cdh2). Yellow outlines in panels (**E**) and (**F**) outline the location of GFP donor-positive tissue. (**G**) and (**H**) Summed slice Z-projections of HH4-HH4 and HH8-HH4 grafts, respectively, at 30 min after grafting stained for DAPI, laminin, and actin. (**I**) Gene Ontology (GO) term analysis of genes differentially expressed in cluster 6 of the dataset represented here as a bar chart where the negative log of the adjusted p-value for each GO term is plotted. The colour of the bar represents the number of genes annotated with that GO term. (**J**) Schematic showing the cells utilised to look for differential expression between HH8 and HH4-derived cells in the central populations (clusters 6, 8, and 12). (**K**) Bar chart showing key GO terms enriched amongst genes differentially expressed between HH8 and HH4 cells in clusters 6, 8, and 12. (**L**) Violin plots showing the expression of the genes Hoxa7 and Hoxb8 in these cells by sample (note that only donor GFP+ cells are included in this plot). Scale bars in (**A'**), (**B'**), and (**E–H**) represent 50 µm. Epi: epiblast; ME: mesendoderm.

The online version of this article includes the following figure supplement(s) for figure 3:

**Figure supplement 1.** Supporting data for single-cell analysis of stage differences.

Taken together, the results from our scRNA-seq experiment show that upon grafting, HH8 cells initially occupy a region of gene expression space that is distinct from HH4 cells, but within 3 hr undergo transcriptomic changes that place them in a state similar to surrounding host tissue, with the exception of *Hox* gene expression differences. These results are therefore consistent with the notion that intrinsic timer(s) govern both *Hox* gene expression and progenitor addition, but that this latter aspect is independent from transcription changes of EMT-associated genes.

## Both HH4 and HH8 cells migrate in an ex vivo culture system but with different dynamics

Having identified that HH8 grafts are delayed in their axis contribution at the level of cell migration from the graft site, we wondered whether this is an outcome of the interaction between the donor and host tissue in this context or intrinsic to the grafted tissue. To test this, we developed an ex vivo culture system where MSP tissue was dissected and cultured on fibronectin-coated glass dishes. We find that mesodermal cells readily migrate as a sheet in this context (*Figure 4—figure supplement 1*). Both HH4 and HH8 MSP tissue migrate on fibronectin (*Figure 4A and C*), so we used particle image velocimetry (PIV) to characterise the dynamics of this process in more depth. Representative explant velocity fields are shown in *Figure 4B and D* for HH4 and HH8 explants, respectively. HH4 explants undergo a rapid expansion in area with no or little lag time before this expansion begins (*Figure 4E*), whereas HH8 tissue does not show a substantial increase in area for the first 10–15 hr of imaging, upon which time it expands relatively slowly (*Figure 4E*). The same trends can be seen in the velocity data, with HH4 explants showing an initial average velocity of 0.2 µm/min and a maximal average velocity of ~0.27 µm/min at 9–10 hr of culture (*Figure 4F*). In contrast, HH8 tissue has a relatively constant velocity of ~0.04 µm/min rising to ~0.1 µm/min at 14–15 hr of culture (*Figure 4F*). Thus, HH8 tissue shows a slower increase in area (and corresponding lower velocity of tissue flow) than HH4 tissue, as well as a substantial lag time before spreading. This suggests that HH8 tissue intrinsically has a weaker proclivity to migrate than HH4 tissue and importantly could account for the lack of early migration we observe in heterochronic HH8 to HH4 grafts.

## Hox gene expression is intrinsically regulated in both grafts and explants

To further explore the intrinsic capability of MSP explants to regulate their developmental timing, we next aimed to elucidate whether in vivo Hox gene expression is also tissue intrinsic. We identified a larger group of *Hox* genes with differential expression in the MSP region at HH4 and HH8 by screening by RT-qPCR and HCR staining (*Figure 5—figure supplement 1*). These genes include *Hoxa2, Hoxa3,* and *Hoxa6*, all of which are not expressed in the HH4 MSP region but expressed in the HH8 MSP region. We performed HCR for *Hoxa2, Hoxa3,* and *Hoxa6* at 3, 6, 12, and 24 hr after grafting to ask whether HH8 tissue maintains expression of these genes upon grafting or downregulates their

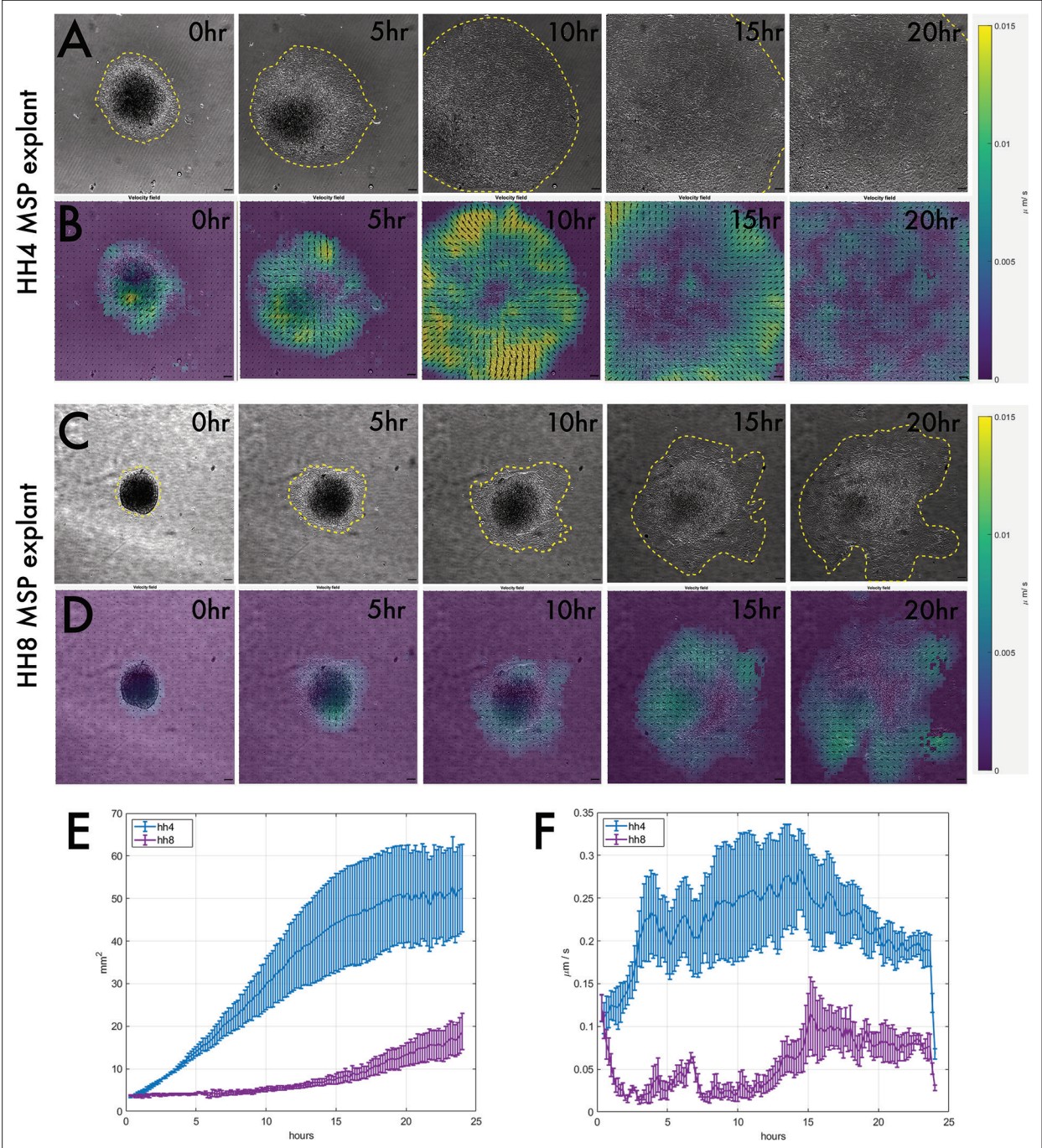

**Figure 4.** Explanting medial somite progenitor (MSP) tissue on fibronectin reveals that HH4 and HH8 MSP tissue intrinsically migrate with different dynamics. (**A**) and (**C**) are brightfield stills taken from timelapse movies of MSP tissue migrating on fibronectin, with (**A**) showing an HH4 explant and (**C**) showing an HH8 explant. Yellow dotted outlines delineate the edge of the migrating explant sheet. (**B**) and (**D**) are velocity field stills for the corresponding images in (**A**) and (**C**), respectively, calculated using particle image velocimetry (PIV). The colour of each interrogation window denotes the velocity of movement (scale on right) between the current time step and the previous one, and the directionality of each vector arrow similarly represents the direction of tissue flow. (**E**) Plot showing the mean area for several explants of each class (n = 5 for HH4 and n = 3 for HH8). (**F**) Plot showing the mean velocity at each timepoint for several explants of each class (n = 5 for HH4 and n = 3 for HH8). In (**E**) and (**F**), error bars represent the standard error. Scale bars in (**A–D**) represent 50 µm.

The online version of this article includes the following figure supplement(s) for figure 4:

**Figure supplement 1.** Characterising migration of HH4 medial somite progenitor (MSP) explants grown on fibronectin.

expression to match the surrounding host environment. We found that invariably HH8 MSP cells retain the donor *Hox* profile upon grafting, suggesting that *Hox* gene expression may be regulated intrinsically at the population level (*Figure 5A–D*). Note that the cell population of interest here is the progenitor population, which is shown in *Figure 5A–D*. We also checked whether heterochronically grafted cells in the formed body axis (i.e. those which have left the progenitor region) express a *Hox* gene complement appropriate for their AP position. We found that both *Hoxa2* and *Hoxa3* are expressed with appropriate domain boundaries in both homochronic and heterochronic grafts (*Figure 5—figure supplement 2*), consistent with an emerging view where cells modify their *Hox* expression profile after exiting the progenitor region by a mechanism independent of the initial activation of gene expression (*Tschopp et al., 2009*).

To investigate the intrinsic regulation of *Hox* expression further, we used explant culture to ascertain whether there is a role for the embryonic environment in progression of *Hox* expression. Explants were made of the HH8 MSP region and cultured for 24 hr as floating explants (*Figure 5E*) before RNA extraction. At 24 hr of culture, explants were maintained as cohesive floating pieces of tissue (*Figure 5F*). RT-qPCR for various *Hox* genes was performed to test for differences between *Hox* expression at 0 and 24 hr after explanting. In each case, this difference was compared to the endogenous change in expression in the MSP region over an equivalent period of time – 24 hr after HH8 embryos reach HH14, so MSP tissue at HH14 was used to represent the endogenous change in expression (*Figure 5G*). Consistent with a progression of the Hox timer in the cells of the explants over 24 hr, we observed an increase or decrease in *Hox* expression consistent with the normal dynamic for each gene surveyed (*Figure 5G*). This data provides support for population-intrinsic regulation of Hox gene expression in MSPs, with this timer able to progress outside of the embryonic environment.

Finally, we performed HH4 to HH8 grafts to ascertain whether there was any asymmetry in the extrinsic regulation of *Hox* gene expression depending on whether tissue was grafted from older to younger or younger to older embryos. We found that in this instance – where HH4 donor tissue does not express *Hoxa2* or *Hoxa7* in vivo, but host HH8 tissue does – there was no evidence for the upregulation of these genes in donor tissue at 6 hr after grafting (*Figure 5H*), further supporting an intrinsic mode of regulation of *Hox* gene expression in this context. Together, the experiments described here suggest that *Hox* gene expression in the MSP population is intrinsically regulated and not subject to reprogramming to match the surrounding host environment.

## Size dependency of graft but not explant behaviour suggests that the interaction of donor and host tissue contributes to graft outcome

Given the striking maintenance of HH8 *Hox* gene expression upon grafting, we wondered whether there is a minimal graft size for maintenance of gene expression. Thus, we performed HH8 to HH4 heterochronic grafts with variable-sized pieces of donor tissue (*Figure 6—figure supplement 1*). We found that at the smallest graft size (eighth size, approximately 6000 $\mu m^2$ initial area), donor cells had spread substantially from the initial graft site within 6 hr (*Figure 6A*), whereas all larger graft sizes remained as a single cluster of cells (*Figure 5B*, example shown here is a quarter-sized graft, approximately 12,500 $\mu m^2$ initial area). Interestingly, in the smallest grafts, individual GFP+ cells could be seen intermixed with wild-type host (GFP) tissue (*Figure 6A, enlarged in lower panel*). Notably, even these isolated HH8 cells still maintained expression of the HH8 *Hox* genes *Hoxa3* and *Hoxa6* (*Figure 6B and C*), further supporting the intrinsic regulation of *Hox* gene expression in these cells. Furthermore, when small HH8 to HH4 grafted embryos are grown for 24 hr, grafted cells are found to occupy more anterior locations compared to large HH8 to HH4 grafts (*Figure 6D*). Larger HH8-HH4 grafts tend to have their most anterior somite contribution in somite 4 or 5, whereas small HH8-HH4 grafts have their most anterior contribution between somites 1 and 3. Importantly, this result suggests that *Hox* gene expression can be uncoupled from cell dispersion and demonstrates that *Hox* gene expression is not the only determinant of the timing of cell allocation to the body axis.

One possible explanation for the enhanced dispersion of HH8 tissue at small graft sizes could be that there is a mechanism intrinsic to population size that can trigger the repression of cell migration above a certain size threshold. Alternatively, it could be due to extrinsic factors involving the interaction of host and donor cells that impacts larger and smaller grafted populations differently. To ask whether the size dependency of dispersion is an intrinsic feature of the HH8 tissue, we explanted HH4 and HH8 MSP tissue of varied sizes on fibronectin. If small HH8 explants can migrate at higher

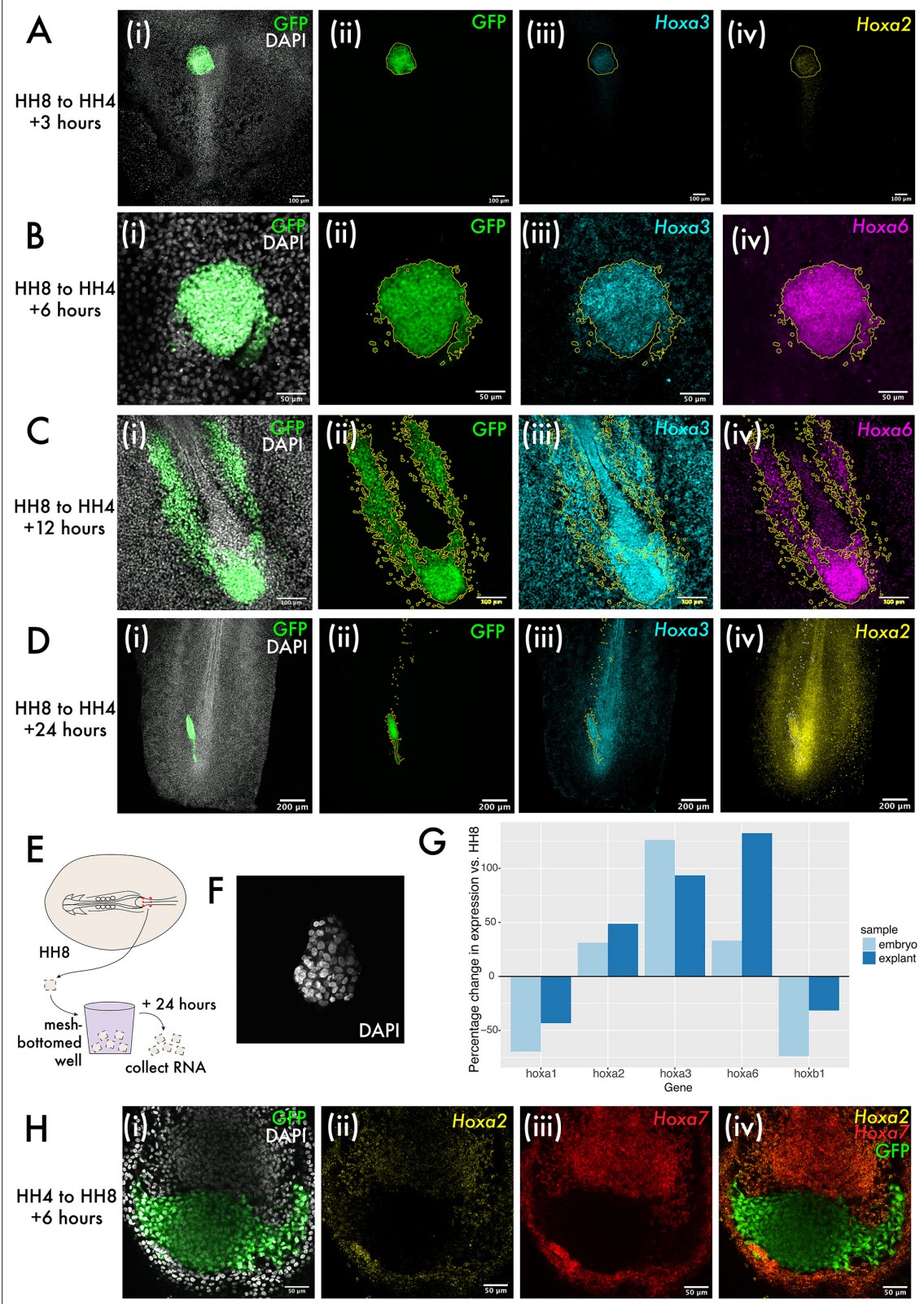

**Figure 5.** Hox expression is intrinsically regulated in both grafts and explants. (**A–D**) show heterochronic HH8 to HH4 grafts at 3, 6, 6, 12, and 24 hr post-grafting, respectively, with each image showing the cells in the progenitor population (tailbud in [**C**] and [**D**]). Panel (**i**) of (**A–D**) shows composite DAPI and GFP summed slice projections. (**ii**), (**iii**), and (**iv**) are the same images as single-channel summed slice projections, showing the location of the donor GFP+ cells, Hoxa3 mRNA and Hoxa2 or Hoxa6 mRNA expression, respectively (please see labels in panel for gene). In (**A–D**), yellow or white lines

*Figure 5 continued on next page*

*Figure 5 continued*

outline the location of GFP+ (donor) tissue. (**E**) Schematic showing explant experiment set-up. (**F**) DAPI stained HH8 explant after 24 hr of culture shown here as a summed slice projection. (**G**) Bar chart showing expression of five different Hox genes in embryonic medial somite progenitor (MSP) regions and HH8 explants after 24 hr, represented relative to the endogenous expression in MSP regions at HH8. (**H**) HH4 to HH8 graft at 6 hr after grafting, stained for Hoxa2 and Hoxa7 expression. (**i**) shows GFP and DAPI fluorescence, (**ii**) shows Hoxa2 expression, (**iii**) shows Hoxa7 expression, and (**iv**) shows Hoxa2, Hoxa7, and GFP. Scale bars in (**D**) represent 200 µm, in (**A**) and (**C**) represent 100 µm, and in (**B**) and (**H**) represent 50 µm. All fluorescence images are presented as summed slice Z-projections.

The online version of this article includes the following figure supplement(s) for figure 5:

**Figure supplement 1.** Hox gene expression characterisation.

**Figure supplement 2.** Hox gene expression in the formed axis of grafted embryos.

velocities than large HH8 explants on fibronectin, this would suggest that HH8 tissue can intrinsically account for the graft outcomes observed. We found that for both HH4 and HH8 tissue, smaller explants exhibit lower velocities of tissue flow and achieve smaller areas throughout the observed 24 hr period (*Figure 6E and F*). This suggests that the enhanced dispersion of small HH8 grafts in the HH4 environment cannot be accounted for by the HH8 donor tissue alone and instead is an outcome of the interaction between HH8 and HH4 tissue. Thus, despite the intrinsically regulated differences in gene expression and migratory dynamics between HH4 and HH8 tissue, extrinsic inputs also influence the timing of cell contribution to the vertebrate primary body axis.

## Discussion
### Mesodermal dispersion is a key step in avian gastrulation

In this work, we first sought to understand at what point in the developmental process of progenitor contribution to the body axis heterochronic grafts of cells from older to younger embryos are delayed. The delayed contribution of axial progenitor cells to the axis that we observe is consistent with previous studies, including mouse tailbud grafts (*Tam and Tan, 1992*) and chicken PS grafts (*Iimura and Pourquié, 2006*). Based on single-cell RNA-sequencing and in vivo gene expression analyses, we propose that MSP contribution to the body axis can be conceptualised as consisting of three successive phases: ingression, dispersion, and migration. In the first step, mesodermal progenitor cells of the PS undergo a classical EMT and ingress from the dorsal cell layer to the ventral one. This EMT involves a well-characterised set of molecular players including the upregulation of *Snail2* (*Slug*) and a switch from E-cadherin (*Cdh1*) to N-cadherin (*Cdh2*) expression (*Cano et al., 2000*; *Dady et al., 2012*; *Nieto et al., 1994*; *Ohkubo and Ozawa, 2004*). Once cells enter the mesendodermal cell layer, substantial and rapid cell dispersion occurs in both the anteroposterior and mediolateral axes. As grafted cell populations become situated within the ventral layer of the embryo as part of the healing process, we are unable to ascertain whether the morphogenetic process of cell ingression is delayed in heterochronic grafts. However, we do observe that the expected changes in EMT gene expression occur within 3 hr of HH8 populations being grafted into HH4 hosts, suggesting that cells are not blocked in this aspect of progenitor cell addition. Instead, cells are delayed in cell dispersion, resulting in the population being maintained as a distinct cluster of cells in the ME and not dispersing readily as control grafts do. These processes are recapitulated in the explant culture system we describe, where both HH4 and HH8 MSP populations ultimately migrate on fibronectin but HH8 populations do so with a delay and at lower velocities. Thus, delayed dispersion of HH8 tissue in heterochronic grafts to HH4 embryos can be accounted for by intrinsic migratory properties of the HH8 tissue.

As discussed in the introduction, the choice of HH4 and HH8 tissue as the focus for this project is interesting in the context of primary versus secondary gastrulation: the occipital somites form simultaneously from tissue that ingresses during primary gastrulation, whereas trunk somites form periodically through the progressive segmentation of a PSM. Here, we have shown that HH4 MSP tissue (pre-node regression) and HH8 MSP tissue (post-node regression) exhibit substantially different migratory dynamics, despite forming a continuous progenitor population. This concept of evolving cell behaviour and properties over time is consistent with the transcriptional changes that have been described in axial progenitor regions (*Wymeersch et al., 2019*). In our scRNA-seq data, we have found that between HH8- and HH4-derived cells of the same cluster, a whole suite of genes are differentially expressed in addition to *Hox* genes including genes associated with the cell cycle, RNA

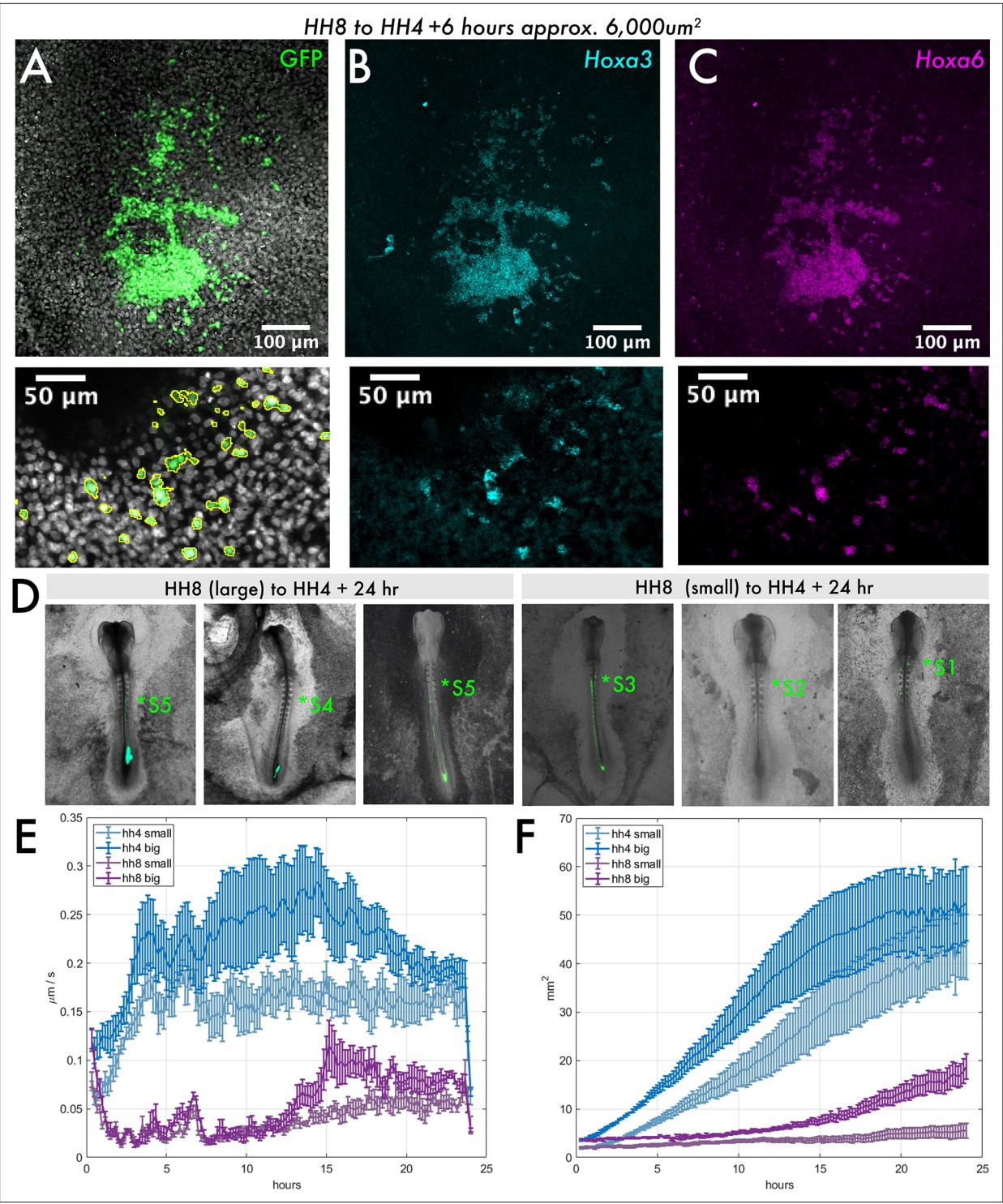

**Figure 6.** Size dependency of HH8 medial somite progenitor (MSP) tissue behaviour. (**A–C**) Small HH8 to HH4 graft (donor tissue ~6000 μm²) at 6 hr after grafting. (**A**) DAPI and GFP composite image, (**B**) Hoxa3 hybridisation chain reaction (HCR) stain, and (**C**) Hoxa6 HCR stain. Lower panels show an enlarged view of images above; GFP signal outlined in (**A**). (**D**) Three examples of HH8-HH4 large and small grafts are shown at 24 hr after grafting as combined GFP/brightfield images. The most anterior somite contribution of each graft is indicated by the annotation (*S). (**E**) Velocity plot showing the mean velocity at each timepoint for several explants of each class (n = 5 for HH4 large, n = 4 for HH4 small, n = 3 for HH8 large, and n = 5 for HH8 small). (**F**) Area plot showing the mean area for several explants of each class at each timepoint (n's as above). In (**E**) and (**F**), error bars represent the standard error. Scale bars in (**D**) represent 200 μm, in (**A–C**) upper panels represent 100 μm, and in lower inset panels represent 50 μm. All fluorescence images are presented as summed slice Z-projections.

*Figure 6 continued on next page*

*Figure 6 continued*

The online version of this article includes the following figure supplement(s) for figure 6:

**Figure supplement 1.** Graft size validation.

splicing, and epigenetic remodelling. Cell cycle differences may reflect differences in the population dynamics over time – the expansion versus depletion of the progenitor population, which is not well-characterised. Some of the genes differentially expressed in the intermediate cell state and associated with adhesion (including *Ptk7* and *Nfasc*) are known to have alternative splice transcripts, raising the possibility that despite not being differentially expressed between HH4 and HH8 cells, differences may arise downstream of transcription (*Hassel et al., 1997*; *Jung et al., 2002*; *Katyal et al., 2007*). A multitude of different cellular processes may be functionally important in the changes in cell behaviour we have described here between HH4 and HH8 tissue, and further research will be necessary to clarify the mechanistic underpinning of the differences we have described.

## A possible role for cell sorting in graft outcome

When HH8 tissue is grafted to a HH4 host embryo, it tends to remain as a single group of cells and minimal mixing is observed with the surrounding host tissue. This is in stark contrast to stage-matched grafts (HH4-HH4), where extensive mixing of donor tissue with host tissue is observed. This result resembles published observations on heterotypic grafts of neural tube tissue from different axial levels in the mouse embryo (*Trainor and Krumlauf, 2000*). Specifically, when homotypic grafts were performed of tissue between the same axial level, extensive cell mixing was observed, but heterotopic grafts did not show this mixing and instead largely remained as a single coherent group of cells (*Trainor and Krumlauf, 2000*). Another study noted that "cells grafted heterochronically integrate less efficiently into host tissues than isochronic grafts (whether heterotopic or homotopic), suggesting that this temporal resetting may be less efficient than their acutely sensitive response to spatial cues" (*Wymeersch et al., 2021*). Together, these observations raise the intriguing possibility that stage-matched tissues can mix more readily than heterotypic or heterochronic grafts because in the latter case donor cells have a greater affinity for other donor cells than host cells, and vice versa. This would resemble the mechanisms of self-recognition-based sorting that have been described in diverse contexts including, for example, *Xenopus* gastrulation (*Winklbauer and Parent, 2017*).

Cell sorting is generally believed to result from adhesion differences between tissues. Early experiments in which three species of sponge were dissociated and reaggregated showed the formation of masses preferentially containing cells of the same species, suggesting that these cells have some ability to recognise others of their own kind (*Wilson, 1907*). Similarly, when cells of amphibian neurulae are disaggregated and allowed to reaggregate, they initially form a mass where any cell type will associate with any other – but within 20 hr fully sort out (*Townes and Holtfreter, 1955*). This was found to be true across a huge variety of tissue combinations, and both when tissues were fully disaggregated or combined as explants (*Townes and Holtfreter, 1955*). More recently it has been suggested that though cell sorting is unlikely to be a major contributor to morphogenesis, these mechanisms may be important for maintaining tissue boundaries (*Winklbauer and Parent, 2017*). Although the differential gene expression analysis between HH4 and HH8 MSPs described here has not highlighted transcripts for any candidate molecules known to modulate adhesion, as mentioned above, it is entirely possible that due to differences at a protein level, the two stages of tissue show preferential adhesion for other cells of the same stage. A precedent exists for a changing adhesion profile over developmental time: such a dynamic process was described in 1939 by Ernest Just in sea urchin embryos (*Just, 1939*). In a future study, it would be interesting to perform pairwise grafts of tissue between embryonic stages and quantify the extent of cell mixing: this quantity would be expected to scale with the extent of similarity between the tissues. Much remains to be understood about the ways that transplanted tissues interact with their host environment and in certain cases show a greater extent of 'incorporation' and mixing with their surrounding host tissue.

## Hox gene regulation is population-intrinsic for somite progenitors

Vertebrate embryos express *Hox* genes with a special dynamic – each paralogous cluster of *Hox* genes is expressed with spatiotemporal collinearity, meaning that they are transcribed in a strict 3' to

5′ order both in space and time. It is known that the location within the cluster defines the timing of expression of each gene, that *Hox* clusters undergo epigenetic remodelling to increase DNA accessibility in a 3′ to 5′ direction, and that the initial induction of Hox gene expression at gastrulation is dependent upon Wnt signalling (*Deschamps and Duboule, 2017*; *Izpisúa-Belmonte et al., 1991*; *Kmita and Duboule, 2003*; *Liu et al., 1999*; *Moreau et al., 2019*; *Neijts et al., 2016*; *Tschopp et al., 2009*). In the context of human pluripotent stem cell-derived axial progenitors of the spinal cord, it has also been shown that progressive Hox activation occurs in vitro, and that this is dependent on an increase in FGF and then GDF signalling (*Mouilleau et al., 2021*). In this work, we find evidence for a population-intrinsic mechanism of *Hox* gene expression within the MSP fated region of the PS: once HH8 cells are expressing a given *Hox* gene, they will maintain this expression profile in a novel HH4 environment. However, the MSP population also expresses FGF ligands (*Bell et al., 2004*; *Darnell et al., 2007*), and it could be that a similar mechanism is at play to explain the population-intrinsic control of *Hox* expression here too. The maintenance of *Hox* gene expression in a new environment (a younger embryo) is consistent with a result reported by *Iimura and Pourquié, 2006*, wherein *Hoxb9* expression was shown to persist at 6 hr after heterochronic grafting of streak cells. However, our results differ from those reported by *McGrew et al., 2008*, who observed reprogramming of *Hox* gene expression in HH15 tailbud chordoneural hinge cells transplanted to the equivalent region of the HH8 embryo – grafted cells no longer express the *Hox* gene *Hoxa10* at 8 hr after grafting. This difference may reflect the different cell populations studied or variation in the contributions of extrinsic and intrinsic influences to *Hox* regulation over the period of primary body axis formation; it is possible that there is a transition from intrinsic to a more extrinsic mode of *Hox* regulation later in primary body axis development. Alternatively, it may be explained by the relative mismatch between donor and host tissue. In the experiments of *McGrew et al., 2008*, *Hoxa10* expression is absent from the progenitor region at HH8-HH10 but expressed at HH15. *Hoxa10* expression is first detected in the progenitor region at HH12 (17 somite stage) (*Denans et al., 2015*), corresponding to a theoretical 20 hr lag time before expression of *Hoxa10* in host tissue of HH8 to HH15 grafts. Conversely, our HH8 to HH4 grafts experience a much shorter lag period before host expression of the *Hox* genes surveyed – expression of *Hoxa2,* for example, is evident in host tissue within 6 hr. Additional grafts and additional *Hox* genes must be studied to ascertain which of these possible explanations is most accurate.

We have seen that the difference in behaviour between homochronic and heterochronic grafts correlates with the persistence of the HH8 *Hox* gene expression profile, raising the possibility that the HH8 *Hox* profile may influence the transition from mesenchyme to invasive mesenchyme. Previous experiments that overexpressed *Hox* genes in PS tissue showed that cells were delayed at the point of ingression through the streak (*Iimura and Pourquié, 2006*). However, because donor tissue healed in the ventral compartment of the embryo after grafting, we were unable to observe the morphogenetic act of ingression and so unable to definitively distinguish between *Hox* regulating ingression or dispersion – functional experiments manipulating *Hox* gene expression would be necessary to make this distinction. Nonetheless, we have demonstrated that *Hox* gene expression can be uncoupled from cell allocation to the axis in small transplants, suggesting that if *Hox* does influence these processes, its effect can be overcome by extrinsic influences under certain conditions.

## Population size-related inputs on cell allocation to the axis

Our experiments show that smaller grafts of older progenitor populations to PS stage embryos do not remain as a single mass but instead show substantial mixing with host tissue, like homochronic grafts. The observation of graft size influencing outcome resembles previous findings that grafts of neural tissue have different outcomes dependent upon graft size (*Couly et al., 1998*; *Guthrie et al., 1992*; *Trainor and Krumlauf, 2000*). A common differential outcome is the maintenance of donor gene expression in larger grafts but not in instances when donor cells are isolated or in smaller groups (*Schilling et al., 2001*; *Trainor and Krumlauf, 2000*). Interestingly, in our experiments, we observe maintenance of the donor *Hox* profile even when cells are not contacting any other donor cells, suggesting that *Hox* gene expression in this context may be truly cell intrinsic. Cell-intrinsic timers have been described in diverse contexts when individual cells are isolated from their developmental contexts, including the somitogenesis clock (*Webb et al., 2016*), the *Drosophila* neuroblast clock (*Grosskortenhaus et al., 2005*), and oligodendrocyte precursor cell (OPC) differentiation timing (*Temple and Raff, 1986*).

In addition to the maintenance of *Hox* gene expression, we also find that smaller heterochronic grafts are able to disperse more readily and contribute to more anterior regions of the body axis. How might graft size influence the timing of cell dispersion and axis contribution? The tendency of smaller HH8 grafts to disperse may be related to tissue properties such as stiffness differences between the HH4 and HH8 progenitor regions. This could make it difficult for HH8 cells to invade the HH4 tissue (and vice versa), leading to little mixing. It also might be possible that due to adhesion differences and differential affinity of HH4 and HH8 tissue for like tissue, that HH8 grafts may undergo some rearrangement (e.g. rounding up to contact other HH8 cells) that slows its dispersion and minimises mixing. Smaller grafts have a relatively large surface area in contact area with surrounding host tissue relative to their volume when compared with larger grafts. There may also be more loose edges of tissue in smaller grafts as a result of the additional tissue cutting involved, creating additional surface area. This may facilitate mixing of HH4 and HH8 cells in smaller heterochronic grafts, allowing HH8 cells to disperse with the surrounding host tissue.

## Both intrinsic and extrinsic cues influence timing of axial progenitor axis contribution

Together, our work has shown that heterochronic somitic progenitor grafts from older to younger embryos are delayed by remaining as a single group of cells rather than mixing with surrounding host tissue. This difference despite cells occupying the ventral compartment of the embryo and correlates with the intrinsic regulation of *Hox* gene expression, such that expression is robustly maintained upon grafting to the HH4 environment. We find that despite this maintenance HH8 cells can disperse if they are transplanted in relatively small groups, suggesting that *Hox* gene expression is not the only input timing cell contribution to the primary body axis. Indeed, our explant culture experiments show that size dependency of migration rates is not evident when cells are cultured in a similar environment, suggesting that the faster contribution of small cell grafts in vivo is due to differences in the ability of donor cells to interact with the host environment. This is likely to be related to differences in the initial distribution of laminin and/or the ability of cells to breakdown the dense structures observed in larger grafted populations (*Figure 2G–H*). Overall, our results are consistent with an emerging view of vertebrate developmental timing where inputs across length scales as well as a combination of intrinsic and extrinsic timer mechanisms give rise to a stereotyped order and timing of events in development (*Busby and Steventon, 2021*; *Chinnaiya et al., 2014*; *Matsuda et al., 2020*; *Rayon et al., 2020*; *Pickering et al., 2018*; *Saiz-Lopez et al., 2015*; *Saiz-Lopez et al., 2017*).

## Materials and methods
### Chicken husbandry

Bovans brown chicken eggs (Henry Stewart & Co., Norfolk) and GFP transgenic chicken eggs (Roslin Institute, Edinburgh) were stored upon arrival at 18°C until use, then incubated at 37°C in a humidified incubator to obtain embryos of an appropriate stage according to *Hambuger and Hamilton, 1951*.

### Tissue fixation

Embryos were fixed for analysis by dissection in PBS (Sigma-Aldrich) and fixation in 4% PFA (Sigma-Aldrich) in PBST either overnight at 4°C or for 1 hr at room temperature (RT).

## Experimental embryology
### Grafting

Homochronic and heterochronic grafts were performed with host embryos in New Culture (New, 1955). Donor GFP-expressing tissue pieces (*McGrew et al., 2008*) containing ~1000–1200 cells were dissected from the embryo in PBS, then transferred to the host embryo in new culture with a glass capillary tube. The graft was carefully positioned at the graft site under PBS (Sigma-Aldrich) immersion using an eyelash tool, and excess PBS removed, before re-incubation at 37°.

### Floating explant culture

Explants of the HH8 MSP region were prepared as follows. Wild-type embryos were removed from the egg, washed in PBS, and the MSP region isolated using a micro-dissecting knife. Explants were cultured in Medium 199 (Gibco) in Millicell cell culture inserts (based on methods in *Streit and Stern, 1999*) for 24 hr, before collection for either fixation or RNA extraction.

### Fibronectin (adhesive) explant culture

Glass-bottomed imaging dishes (Mattek) were coated with human fibronectin protein (1 mg/mL stock solution diluted 1:40 in PBS) by pipetting the solution onto the dish and incubation at 37°C for 1–3 hr. Fibronectin solution was removed, and the dish allowed to dry for 15 min at 37°C. Explants of the HH4 or HH8 MSP region were prepared as follows. Wild-type embryos were removed from the egg, washed in PBS, and the MSP region isolated using a micro-dissecting knife. Explants were placed on the fibronectin coated dish and oriented mesendoderm (ventral) side down using an eyelash tool and cultured in GMEM (Gibco) + 10% foetal bovine serum (FBS) + 1% Pen/Strep.

## Molecular biology

### Hybridisation chain reaction (HCR)

RNA expression was visualised using third-generation HCR (*Choi et al., 2018*). Embryos were rehydrated through a descending methanol series, washed twice in PBST, then digested in Proteinase K (10 μg/mL) for 5–10 min at RT. Embryos were then incubated with 4% PFA in PBST for 20 min, washed twice in 5× SSCT for 10 min each, and then incubated in HCR hybridisation buffer (Molecular Instruments) for 30 min at 37°C. HCR probes (Molecular Instruments) were added to hybridisation buffer at 4 $\rho$ mol/mL, and embryos were allowed to incubate in this probe solution overnight at 37°C. The next day, embryos were washed four times for 15 min in HCR wash buffer (Molecular Instruments) at 37°C, then washed twice in 5× SSCT at RT for 10 min. Embryos were incubated in HCR amplification buffer (Molecular Instruments) for 5 min. Fluorescently labelled HCR hairpins (Molecular Instruments) were snap-cooled by incubation at 95°C for 90 s before cooling to RT in darkness. HCR hairpins were added to HCR amplification buffer at 60 $\rho$ mol/mL each, then embryos were incubated in this hairpin solution overnight at RT in darkness. Hairpin solution was removed, and embryos were washed in 5× SSCT for 10 min three times at RT. DAPI staining was performed by incubation of embryos in DAPI (1:1000) in 5× SSCT for 30 min at RT. Finally, DAPI solution was washed off by three 10 min washes in 5× SSCT at RT.

### Immunohistochemistry

Embryos were blocked for 1 hr at RT in blocking buffer (2% normal goat serum [Sigma-Aldrich] in PBDT). Primary antibody (Laminin, Sigma L9393) was added to blocking buffer at 1:25 and embryos were incubated in antibody solution overnight at 4°C with agitation. The following day, embryos were washed six times for 20 min in PBDT at RT, before addition of secondary antibody (1:1000) and DAPI (1:1000) in blocking buffer overnight at 4°C. Unbound secondary antibody was removed by washing in PBDT six times for 20 min at RT.

### Phalloidin staining

Fixed embryos were stained for F-actin using Alexa 568 conjugated phallodin (Thermo Fisher). Embryos were incubated in a 1:1000 dilution of phallodin in PBS for 1–3 hr at RT before washing in PBS.

### RNA extractions

Tissue was dissected in cold PBS and transferred to 1 mL cold Trizol (Invitrogen), homogenised using a pestle, and allowed to incubate at RT for 5 min. Then, 200 μL chloroform was added and the mixture was incubated for 3 min at RT. Samples were centrifuged for 15 min at 4°C, and the upper aqueous phase was retained and transferred to a new tube. Then, 500 μL isopropanol was added to the aqueous phase, incubated for 10 min at RT, and centrifuged for 10 min at 4°C. The supernatant was discarded, and the pellet washed with 75% ethanol, before resuspension in RNase-free water and incubation at 55°C for 15 min.

## RT-qPCR

RNA was reverse transcribed to cDNA using SuperScript III reverse transcriptase at 50°C for 60 min. RNA was removed by RNase H digestion for 20 min at 37°C. QPCR was performed using the QIAGEN Rotor-Gene Q machine, with SYBR Green Mastermix. Reactions were run in triplicate. Gene expression was quantified by standard curve analyses using serial dilution series of HH7 whole embryo cDNA. The threshold cycle (Ct) was calculated for each dilution in the series (using a threshold of 0.02) and used to produce a log dilution-Ct graph for each gene, allowing calculation of the relative concentration of transcript in each unknown sample. Primer sequences are available in *Supplementary file 1d*.

## Single-cell RNA-sequencing

Samples were prepared for single-cell RNA sequencing as follows: tissue was dissected in cold PBS and washed thrice in cold PBS. PBS was replaced with dissociation solution (Trypsin + 0.05% EDTA, Gibco) and samples were incubated at 37°C for 15 min, with trituration every 5 min. The dissociation reaction was terminated by addition of 10% FBS. Samples were spun down at 4°C for 5 min and the supernatant discarded. Each cell pellet was resuspended in PBS + 0.25% BSA and passed through a 40 μm pluriStrainer cell strainer (pluriSelect). cDNA libraries were prepared using the 10X Genomics 3'mRNA-seq workflow (10X Genomics) and sequenced using a NovaSeq instrument (Illumina). Data was processed using CellRanger (10X Genomics) to align reads to a reference transcriptome (produced using the galGal6 genome assembly published by Genome Reference Consortium) and produce matrices of gene expression. Matrices were then passed into Seurat in R *Satija et al., 2015* to perform clustering analyses and identify differential gene expression amongst clusters. Gene Ontology analysis was performed with the GeneCodis webtool (*Carmona-Saez et al., 2007*).

## Imaging

Live embryos were imaged on a fluorescent dissecting scope (Leica) in brightfield and GFP channels. Images were overlaid using GNU Image Manipulation Programme (GIMP) to produce the composite images presented in the article. Confocal imaging of fixed and stained embryos was performed using a Zeiss LSM 700. For confocal imaging, embryos were mounted in Vectashield between two coverslips separated by double-sided tape and sealed using clear nail polish. For the live imaging of adherent explants, a Nikon widefield scope was used with a heated chamber (37°C).

## Particle image velocimetry (PIV)

The velocity fields at the surface of the explants were computed by digital PIV using PIVLab vs2.56 for MATLAB with two passes of 64 × 64 pixels and 32 × 32 pixels with a 50% overlap. To reduce the noise and remove small fluctuations, each interrogation window was averaged with its 7 × 7 neighbourhood, and with the same location in the previous and following timepoint. Velocity vectors outside of the explant were eliminated using a mask generated using Trainable Weka Segmentation, a Fiji plugin that allows training of machine learning models for per-pixel classification (*Schindelin et al., 2012*). The pixel size for this data is 1.29 μm and the time step is 10 min.

## Acknowledgements

LB was supported by a BBSRC DTP Studentship, GSN was supported by a research grant from the Leverhulme Trust (RG93881), and BS was supported by a Henry Dale Fellowship jointly funded by the Wellcome Trust and the Royal Society (109408/Z/15/Z). We thank Alexandra Neaverson, Val Wilson, Stephane Nédélec, and Kate McDole for valuable feedback.

## Additional information

### Funding

| Funder | Grant reference number | Author |
| --- | --- | --- |
| Wellcome Trust | 10.35802/109408 | Benjamin John Steventon |

| Funder | Grant reference number | Author |
|---|---|---|
| Leverhulme Trust | RG93881 | Guillermo Serrano Nájera |
| Biotechnology and Biological Sciences Research Council | Studentship | Lara Busby |
| Wellcome Trust | 225360/Z/22/Z | Benjamin John Steventon |

The funders had no role in study design, data collection and interpretation, or the decision to submit the work for publication. For the purpose of Open Access, the authors have applied a CC BY public copyright license to any Author Accepted Manuscript version arising from this submission.

## Author contributions

Lara Busby, Conceptualization, Data curation, Formal analysis, Validation, Investigation, Methodology, Writing – original draft, Writing – review and editing; Guillermo Serrano Nájera, Formal analysis, Visualization; Benjamin John Steventon, Conceptualization, Supervision, Funding acquisition, Project administration, Writing – review and editing

## Author ORCIDs

Lara Busby http://orcid.org/0000-0003-0705-1364
Guillermo Serrano Nájera http://orcid.org/0000-0001-5841-8408
Benjamin John Steventon https://orcid.org/0000-0001-7838-839X

## Decision letter and Author response

Decision letter https://doi.org/10.7554/eLife.90499.sa1
Author response https://doi.org/10.7554/eLife.90499.sa2

# Additional files

## Supplementary files

• MDAR checklist

• Supplementary file 1. Supporting differential gene expression and primer sequences. (a) Differential gene expression (clusters 0–13). Differential expression analysis was performed in Seurat to identify 'markers' of each cluster, that is, those genes whose expression differed significantly between a given cluster and the remainder of the dataset. For each cluster, the 10s genes with the most significant (smallest) adjusted p-value are provided in this table. (b) GO term analysis for cluster 6. GeneCodis was used to perform Gene Ontology analysis on the full list of differentially expressed genes in cluster 6 conserved across samples (i.e. genes differentially expressed in cluster 6 across all three samples). (c) Differential gene expression for HH8- and HH4-derived cells in clusters 6, 8, and 12. Pooled HH8- and HH4-derived cells in the central clusters of the UMAP plot were used to test for genes differentially expressed between these groups. Cell cycle-related genes were removed from the analysis. (d) Primer sequences for RT-qPCR.

## Data availability

The single cell RNA-sequencing dataset generated in this work is available via the NCBI Gene ExpressionOmnibus (GEO) under the accession code GSE224169.

The following dataset was generated:

| Author(s) | Year | Dataset title | Dataset URL | Database and Identifier |
|---|---|---|---|---|
| Steventon Ben | 2023 | Gene expression of dissected portions of the chicken embryo gastrula after homochronic and heterochronic grafts of somite progenitors | https://www.ncbi.nlm.nih.gov/geo/query/acc.cgi?acc=GSE224169 | NCBI Gene Expression Omnibus, GSE224169 |

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
