## [Editor Report]

This interesting study provides valuable data exploring how progenitors control their contribution to somitogenesis. By combining classic embryology techniques with single-cell sequencing, the authors describe novel cell states that might help understand progenitor population dynamics. This manuscript will be of interest to researchers in the development field who want to better understand hox control and influence during axial elongation.

---

## [Decision Letter]

[Editors' note: this paper was reviewed by Review Commons.]

---

## [Author Response]

**General statements**

We thank all three reviewers for their thoughtful comments and suggestions and believe that the manuscript is stronger for incorporating and addressing them.

A control HH8 to HH8 homochronic graft to check the behavior of the grafted cells: do they disperse in their natural environment after ingression through the primitive streak or they are also paused as a distinctive cell sluster?

We have performed HH8 to HH8 homochronic grafts as suggested and have included images of these grafts at 24 hours as part of modified *Figure 1* (*panels R, S, S’*). We find that homochronic HH8 to HH8 grafted tissue (GFP-positive) shows substantial mixing with surrounding host tissue (GFP-negative). Additionally, the anteroposterior contribution of these grafts is more posterior than the contribution of HH8 to HH4 heterochronic grafts (the most anterior somite contribution is more posterior than somite 4/5), suggesting that while HH8 tissue is delayed in axis contribution relative to HH4 tissue in the host environment, it nonetheless has a substantially more anterior contribution relative to HH8-to-HH8 stage-matched grafts. This result suggests that while tissue intrinsic cues influence axis contribution timing, extrinsic cues that vary between the HH4 and HH8 environment must also play a role. We have added text making this point to accompany the edited figure (*lines 143-150*).

- The reverse heterochronic grafting experiment, namely HH4 cells into HH8. Do HH4 cells maintain their dispersal behavior at the ectopic position, or they behave differently?While the authors assessed the intrinsic properties of HH4 and HH8 tissue by incubating it on fibronectin, this experiment does not properly reproduce the environment of the embryonic region receiving the graft, which might be different at HH4 and HH8. The experiments I am suggesting take this variable into consideration and will therefore help assessing the possible involvement of the host tissue in the behavior of the grafts.

The original manuscript included the reciprocal graft (HH4 in HH8 environment) at 6 hours after grafting in Figure 5I (now Figure 5H), in the context of assaying *Hox* gene expression in this graft type. It is difficult to quantify ‘dispersion’ per se in the bounded compartment of the PSM (versus the unbounded mesendodermal compartment present at HH4) but there does appear to be movement of cells from the graft site at 6 hours after grafting (Figure 5H). Please note that the HH4 to HH8 graft is particularly technically difficult and while we performed this experiment numerous times, we were unable to recover sufficient numbers of embryos to confidently draw conclusions regarding somite contributions of these grafts. Regarding the question of host tissue involvement, please see the new *Figure 6* that addresses this issue.

In addition to those experiments, a more extensive analyses of the already reported experiments could also improve the manuscript.1. When the cells staying in the MSP region after homochronic HH4 grafts reach later stages (e.g. approaching HH8), do they keep dispersing as at earlier stages after ingression through the primitive streak or they remain as a distinct cluster?And does Hox gene expression within those grafts follow the same activation profile observed in the host cells as development proceeds?

Our original manuscript contains images of homochronic HH4 to HH4 grafts cultured to timepoints after HH8, where somite contribution of these grafts continues posterior to the most anterior 4 somites (those formed by HH8) and donor cells can be seen intermixed with host cells in the PSM (Figure 1G’’). This result would suggest that the donor cells do not become distinct from their host tissue at HH8 in the case of homochronic grafts.

Regarding *Hox* gene expression within grafted tissue, we have stained homochronic HH4-HH4 grafts for *Hox* genes at 24 hours after grafting (Figure 5 Supplement 2). At 24 hours after grafting, GFP-expressing donor tissue in the somites and in the tailbud region expresses *Hox* genes appropriate for its context. *Hoxa2* is expressed in somites posterior to and inclusive of somite 1, whilst *Hoxa3* transcripts are present in somites posterior to and including somite 4 (Figure 5 Supplement 2). These boundaries are present in the GFP-positive donor tissue and congruent with expression in the surrounding host tissue (Figure 5 Supplement 2). Similarly, *Hoxa2* and *Hoxa3* are expressed in the tailbud of HH10 embryos (old Supplementary Figure 5, new Figure 5 Supplement 1). Donor cells in the presomitic mesoderm of embryos 24 hours after heterochronic grafting express *Hoxa3* like surrounding host tissue. Together, these data support the normal progression of *Hox* gene expression in homochronically grafted tissue.

2. In the experiment reported on Figure 5I, HH4 MSP grafted into HH8 embryos fail to activate Hox genes like Hoxa2, even after 6 hours of incubation. When these grafted embryos develop even further (for the period of time required for a HH4 embryo to reach the HH8 stage), do they activate Hoxa2 or Hoxa3 or they remain negative for these genes?

As discussed above, we have been unable to recover sufficient numbers of HH4 to HH8 grafts to determine the timing of upregulation (or absence of upregulation) of *Hox* genes in the grafted tissue. The reciprocal graft was not the major focus of this work and so while an interesting question, we believe that this experiment falls outside the scope of the work.

3. The differential GO terms between HH4 and HH8 tissue in cluster 6 include chromatin organization, DNA methylation and C5-methylation of cytosine. This suggests that epigenetic changes might be involved in the behavioral differences between the MSP of the two stages, which can affect many different processes involved in cell activity, including the activation of Hox genes.

We thank the reviewer for bringing our attention to this interesting point and agree that this differential expression is important given the epigenetic remodelling previously described in the course of *Hox* gene expression progression. We have added a reference to this in the manuscript that can be found at lines 224-227.

Some comments on data interpretation.1. It is clear that Hox gene expression in the grafts matches the profile of the donor tissue, indicating the existence of a Hox "timer". However, in my opinion, the authors place too much emphasis on the possible meaning of these observations in what concerns the differential behavior of the grafted cells. If they want to focus on Hox genes they should include some experiment testing their involvement in cell dispersal, either by misexpression or downregulation of specific genes (although there is plenty of information arguing against this possibility, maybe with the exception of that of Iimura and Pourquie, 2006; in this regard, the authors' own data already indicate that dispersion is independent of Hox gene expression).

We thank the reviewer for this comment; we do not rule out the possible existence of two parallel but independent timers, one controlling Hox gene expression and one controlling cell behaviour.

In addition, in response to reviewer 2 (major comment 5), we have added a new figure (new Figure 6) that addresses an additional role for donor-host tissue interaction in modulating progenitor addition and modified our conclusions as well as the title of the paper (now ‘Intrinsic and extrinsic cues time somite progenitor contribution to the vertebrate primary body axis’) accordingly.

2. The authors disregard the involvement of differential patterns of cell adhesion molecules as the origin of the differential behavior of HH4 and HH8 grafts in the HH4 context. However, in their data on supplementary Figure 2A there are several genes differentially expressed between the HH8 and HH4 cells (e.g. Ptk7, Spon1 or Nfasc) that could indeed play a role in the differential interaction between cells from the two embryonic stages. It might be interesting to perform HCR experiments with some of these factors to see if they are differentially expressed at the two embryonic stages. Also, although it might be somewhat far reaching, if differential expression is observed by HCR, it might be interesting to experimentally manipulate expression of the relevant gene (s) (misexpression or down-regulation, depending on the stage) to evaluate its/their potential functional relevance.

We completely agree that there is a possible role for cell adhesion in the differential behaviour of cells (though there is no statistically significant difference in expression level of the genes mentioned between HH4 and HH8 cells) and have amended the text to make this clear. It is of course possible that there are adhesion differences that are not evident at the transcriptomic level. Please see lines 360-390 for a treatment of this issue.

Minor points1. The authors write that the HH8 specific clusters are 0, 4 and 6. However, I think that it is #7 and not #6 the one belonging to this group. I guess that this is typo, but becomes confusing, as a large part of the analysis of the single cell data is centered on cluster #6.

We thank the reviewer for bringing this to our attention – this is indeed a typo, and the text has been amended (line 170).

2. In the introduction the authors state that the first 4-5 somites do not develop ganglia, citing Lim et al. 1987. I think that the way this is written is imprecise, as it sort of implies that more caudal somites develop ganglia (which would mean that the dorsal root ganglia are somite derivatives). However, somites at any level do not develop ganglia; the anterior half of their sclerotomes are permissive to migration of the neural crest that will eventually build the ganglia, something that seems not to happen in the more anterior somites.

We agree that this statement is ambiguous and have edited the text to make it clear that the caudal somites are invaded by neural crest rather than differentiating to ganglia (line 88).

A side noteDifferent alternative transcripts have been reported at least for Ptk7 and Nfasc. This might be relevant considering that another of the prominent differential GO terms identified in supplementary Figure 2C is related to RNA splicing. Would different alternative transcripts for some of these genes be specifically associated with the cells from one of the embryonic stages?

This is an interesting comment and we have added a mention of this possibility to the discussion of the manuscript, which can be found at lines 348-358*.*

Significance:It has been known for many decades that the first 4-5 somites of amniotes are different to the rest of the somites in several ways, from the structures they generate to the way they are generated or the gene regulatory networks controlling their morphogenesis. Much is known about how the posterior somites are generated and the mechanisms of their differentiation. Conversely, relatively little is known about the same processes in the most anterior somites. The work described in this manuscript shows that the progenitor cells from the epiblast that will contribute to the 4-5 first somites already behave different than those generating more caudal somites. Also, they show that progenitors generating more caudal somites are unable to contribute to the rostral somites. These two sets of observations show that the differences in the rostral and caudal somites are already present in their progenitors and that those features are quite stable within the cells, at least when they are kept as a group.So far, the single cell analyses shown in this manuscript failed to provide clear hints to explain the different behavior of the two sets of progenitors. However, they represent an important resource to further explore this important biological question. The authors focus on Hox genes as potential regulators of the differential behavior of the HH4 and HH8 MSPs, I guess that prompted by the report by Iimura and Pourquie (2006) indicating the involvement of Hox genes in the migratory properties of the somite progenitors. However, there is plenty of information, mostly genetic studies in mice, indicating that Hox genes might have very little influence in the differential behavior of rostral and caudal somites. In this regard, expression does not mean causation.I think that this manuscript is interesting, most particularly for developmental biologists involved in understanding the mechanisms governing the basic layout of the vertebrate body plan. Research in my laboratory also explores this type of biological questions, although using more genetic approaches and in a different model system, namely the mouse. I therefore consider myself in a position that allows a knowledgeable evaluation of this manuscript.Reviewer 2Evidence, reproducibility and clarityIn this manuscript, the authors investigate the importance of intrinsic and extrinsic factors in the timing of progenitor addition to the elongating primary body axis. During development, progenitor populations have to combine their selfrenewal with the gradual contribution to the full length of the body axis. The mechanisms underlying the population dynamics that ensure the formation of a proportioned body plan remain poorly understood. By combining heterochronic (HH8 to HH4) and homochronic grafting (HH4 to HH4) of somitic progenitors with next generation sequencing and imaging, the authors observe that the older HH8 tissue shows intrinsic delays in migration and does not disperse within the surrounding mesodermal tissue after ingression through the primitive streak. This behavior correlates with intrinsic and tissue-specific differences in the expression of Hox genes but not with differences in the expression of cell adhesion/migration genes.Overall, this study provides new data exploring how progenitors control their contribution to the body axis. By combining classic embryology techniques with single-cell sequencing, the authors describe novel cell states that might help understand the progenitor population dynamics. There are however a number of further analyses and experiments that should be performed to support the main claims of the manuscript.Major comments:1. The authors claim that grafted HH8 cells are paused after the ingression stage and before the dispersion stage. The grafted cells ingress through the primitive streak and then remain as a distinct cluster of cells that does not disperse throughout the mesoderm. This is in contrast with other obs ervations where overexpression of late hox genes delays the cells at the point of ingression. The authors should better demonstrate that their grafts are actually ingressing and then stopping once in the mesoderm compartment. Figure 3B' shows grafted HH8 cells (GFP positive) present in the mesoderm (ME) compartment 3 h after grafting. It is surprising that a cluster of cells can ingress through the primitive streak in a short period of time and then remain paused. It would be helpful to have the equivalent figure right after grafting to assess the differences in the location of the HH8 GFP+ cells and potentially observe them while ingressing.

This is a good point that we have addressed by characterising HH4-HH4 and HH8-HH4 grafts immediately after the graft has incorporated (~20-30 minutes after grafting). These grafts were fixed and stained with DAPI and phalloidin as well as immunostained for the basement membrane component laminin. We have included this data in the modified manuscript as the new Figure 3G-H.

To summarise our observations, homochronic and heterochronic grafts at 20-30 minutes after grafting can both be found ventrally in the embryo. There are differences in the tissue organization of the two graft types including a higher nuclear density in HH8 tissue relative to HH4 tissue and are delineated from surrounding host tissue by a circumferential ring of F-actin. The observation that HH8 to HH4 grafted tissue is found ventrally (below the host basement membrane) at such an early point after grafting suggests that HH8 tissue may not be actively ingressing through the primitive streak initially but is deposited ventrally by the healing of the host tissue. This changes the conclusions of our experiment from cells undergoing ingression through the primitive streak and then pausing before their dispersion to experiencing a delay before undergoing EMT. We have amended the text of the manuscript accordingly to reflect this new interpretation in the light of the new data (lines*198-209*) and thank the reviewer for their suggestion which has allowed us to clarify such an important point.

The authors describe a novel transcription state, namely clusters 6, 12 and 8 in Figure 2B, populated by HH8 cells 3 h after grafting. It is surprising that the UMAP looks very different between 0 h and 3 h in the HH8-HH4 grafts (Figure 2E and F). The authors should clarify where the HH4 (GFP negative) cells are present in Figure F, I. In the current figures, it looks as if both HH8 and HH4 cells changed completely their transcription profile in only 3 h and populated the central clusters (6, 12, 8). The authors claim that these central clusters are present in normal development and that cells rapidly transit through them. However, it is not clear whether this state happens before or after HH4. For example, the cells may be moving from right to left in UMAP_1 according to time (HH4 in the right, HH8 in the left and a central transient cluster). This would mean that in Figure 2F HH8 grafted cells are regressing to an earlier development state and not a new one. Including RNA velocity analysis could help clarify how the cells are changing their expression profiles.

From data present in the original manuscript, the HH4-HH4 homochronic graft sample at 0 hours has both donor and host cells present in the central set of clusters (Figure 2G), suggesting that HH4 cells transit through this cell state. There is a caveat to this experiment in that we have not included a sample of ungrafted tissue. The reviewer states that “it is not clear whether this state happens before or after HH4” – given that this is a sample of HH4 cells this is not the case, and the cell state therefore is present in the embryo at HH4. In addition, given that HH8 grafted cells at 3 hours maintain *Hox* gene expression, we do not think it correct to say that HH8 cells are “regressing to an earlier development state”.

One factor which should be mentioned is that there are a smaller number of cells in the HH8-HH4 3 hours sample than the others leading to a sparser population of the UMAP plot, which may be leading to difficulty interpreting the location of cells. We have replaced the original UMAP plot with plots showing just the *egfp-*negative cells and just the *egfp-*positive cells (Figure 2G) to make the representation of these classes of cells in each cluster clearer to the reader. In these plots, we show that clusters 6,12 and 8 are indeed present in host HH4 tissue (*eGFP*-negative) in all three datasets, supporting this notion that this is a gene expression state observed in normal development.

4. Related to the previous point, the striking changes between 0 h and 3 h in the HH8-HH4 grafts (Figure 2E and F) may suggest an effect of the grafting procedure on the transcription profile of the cells. The authors should demonstrate that the grafting of cells does not have a huge impact on the transcriptome and that these changes are specific to the previously undescribed delayed state of HH8 cells. For this, they should include scRNA data of HH4-HH4 3 h. If grafting does not have a significant effect on the transcriptome, they should see GFP positive and negative cells in HH4-HH4 3 h remain intermixed.

A comparison of HH4 to HH4 host and donor cells at 3 hours post grafting is now included as Figure 2 Supplement 2. Host and donor cells (distinguished by eGFP expression, Figure 2 Supplement 2A) remain intermixed after clustering and exhibit similar gene expression, consistent with grafting not impacting the transcriptome. We mention this result in the revised manuscript in lines 176-178*.*

5. The authors observe that when doing smaller heterochronic grafts, cells can disperse throughout the mesoderm.Nevertheless, the Hox gene expression does not change depending on the size of the grafts. This is in sharp contrast with their observations and claims done for big heterochronic grafts. The result is interesting as it demonstrates that the expression of Hox genes, but not dispersion, is cell intrinsic. However, the uncoupling of hox gene expression and cell dispersion requires further investigation. The authors should repeat the heterochronic grafting of Figure 1 using smaller grafts and check the contribution of grafted cells to the somites. If cells can readily disperse without delay, they might be able to contribute to all somites as observed with homochronic grafts.

This is an important point that we thank the reviewer for raising. We have performed small heterochronic HH8 to HH4 grafts and allowed them to grow to 24 hours to assay the somite contribution of the transplanted cells. This data is included in the manuscript as part of the new Figure 6. Interestingly, while large heterochronic grafts contribute most anteriorly to somite 4 or 5, our small heterochronic grafts contributed most anteriorly to somite 1, 2 or 3. This is an important result and suggests that in small heterochronic grafts, the early dispersion that we observe leads to a more anterior somite contribution (earlier in time), showing that *Hox* gene expression can truly be uncoupled from AP axis contribution. This result is discussed in the revised manuscript at lines 299-304.

Similarly, the authors should repeat the explant spreading assay using smaller HH8 grafts and quantify whether differences in the migratory dynamics are observed. The authors already discuss the possibility that other factors apart from Hox expression might affect dispersion. Nevertheless, they should assess the importance of graft size in their experimental system.

We have performed the explant experiment with HH4 and HH8 tissue of varying sizes and have included this in the revised manuscript as part of the new Figure 6. The prediction, if enhanced dispersion and earlier somite contribution of small HH8 to HH4 grafts is an intrinsic feature of the smaller piece of donor tissue, would be that small HH8 explants would be expected to spread more rapidly on fibronectin than large HH8 explants, possibly reaching similar migration velocities to HH4 explants. However, we find that migration velocities and area changes in smaller HH8 explants are actually lower than in larger HH8 explants, precluding this explanation for our data. We can thus interpret our results as suggesting that the enhanced dispersion of small HH8 grafts results from the context of grafting and possibly through differences in the interaction of small and large HH8 grafts with the surrounding HH4 tissue. We have included these ideas in the discussion of the new manuscript (lines 447-457). We thank the reviewer for leading us to this finding that highlights the importance of extrinsic factors in timing axis contribution of progenitors. It has led us to an interpretation of our data as supporting a model whereby both intrinsic and extrinsic cues modulate the timing of axis contribution, reflected by the new title “Intrinsic and extrinsic cues time somite progenitor contribution to the vertebrate primary body axis”.

Minor comments:6. It would be informative to have a better time-resolved description of the heterochronic graft behavior in Figure 1. For homochronic grafts, several timepoints are provided allowing the visualization of cells travelling through the body axis. For heterochronic grafts, by contrast, only an early and final timepoint are provided.

To address this comment, we have included images of HH8 to HH4 grafts at several intermediate timepoints after grafting (between 0 and 24 hours) to better characterise their behaviour over this time period and included this data in the revised manuscript as the new Figure 1 Supplement 1.

In Figure 4, the authors show that explants of HH4 and HH8 embryos have different migratory dynamics, with HH4 cells migrating faster than HH8. In Figure 4E, HH8 explants seem not to change their area for about 15 h and then start spreading. This indicates that there is a great delay in migration compared to HH4 explants. However, once they start spreading, it seems that the area starts to increase exponentially in a similar manner to what is observed for HH4 at earlier time points. It would be interesting to monitor the HH8 for a longer time to see the behaviors at later time points. HH8 explants may be just delayed, and once they start fully spreading, the speed may not be so different from the one of HH4 explants.

In the original manuscript, we have shown that the velocity of HH8 explants is relatively stable after 15 hours at values ½ of that seen in HH4 explants (please refer to Figure 4F). For this reason, we do not believe that the reduced spreading of HH8 explants represents a delay relative to HH4 explants (if the velocity were increasing over time this would be a possibility).

7. The authors conclude in Supp. Table 2 that HH8 and HH4 do not have different expressions of adhesion-related genes upon grafting. This observation is very important to understand the potential mechanism behind the different dispersion behaviors, and thus it should be included in the main figure.

We appreciate this point and have moved the GO term bar chart to Figure 3K.

Reviewer #2 (Significance (Required)):The authors combine classic embryology with single-cell RNA-seq and imaging techniques to explore progenitor population dynamics during addition to the body axis. They conclude that the delayed contribution of older cells to axis formation correlates with the intrinsic expression of posterior hox genes. While the idea of intrinsic regulation of hox genes during axial specification is not conceptually new, the authors use modern techniques to describe with finer detail the progenitor population states. For this reason, this manuscript will be of interest to researchers in the development field who want to better understand the hox control and influence during axial elongation.Reviewer #3 (Evidence, reproducibility and clarity (Required)):In this manuscript entitled "A population instrinsic timer controls Hox gene expression and cell dispersion during progenitor addition to the body axis", Busby and colleagues investigate the topic of "cell type identity" in the context of body axis elongation in chick embryos.To this end, they performed heterochronic grafts from HH8 stage embryos to HH4 stage embryos and compared these to HH4 homochronic grafts. They found that HH8 grafts ingressed but were then delayed at a stage they termed cell dispersion. By scRNAseq this new cell state was characterized further. While HH8 cells adjusted their expression pattern to their surroundings, Hox gene expression was maintained as in the host developmental stage. Hox gene expression and collinearity of expression changes were also maintained when HH4 cells were grafted into HH8 embryos or cells were cultured ex vivo. Finally, the authors found differences in migration properties between HH4 and HH8 cells, when cultured ex vivo, with HH4 cells migrating faster than HH8.

This constitutes an elegant work to describe the existence of a "cell-intrinsic timer" that regulates cell identity and progressive body axis extension. Experiments and analysis have been performed adequately and conclusions have been drawn appropriately.There are a rather minor comments I would suggest for further analysis, discussion and potentially experiments to further support this paper:- A major finding is that grafted cells keep their Hox expression pattern, independent of whether it is from HH4 to HH8 or vice versa. Moreover, grafted HH8 cells pause at the cell dispersion stage and do not mix, unless grafted in very low cell populations. The authors conclude that Hox gene expression seems to be cell intrinsically regulated. However, for pausing of cells after ingression, I wonder if it is rather the difference to the neighbors than a cell-intrinsic effect that prevents the cells from dispersing.

We thank the reviewer for their thoughts. We agree that this is important to consider, and the question of intrinsic vs. extrinsic (neighbour interactions with host tissue, for example) factors in causing delay of graft spreading motivated our experiment with small explants (please see above in response to reviewer 2). In light of this data, we agree that it is absolutely important to consider the difference between the cells and their neighbours in the grafted context. We have adjusted our title and conclusions accordingly.

One possibility is that differences in adhesion could account for this, since sorting of cell populations based on differential expression of adhesion molecules has been observed in various model systems. This possibility is excluded here, since adhesion-related genes were not differentially expressed in their expression data. However, I would not exclude this possibility at this stage for the following reasons: 1. The authors detect different migration speeds for HH4 and HH8 cell clusters with HH4 cells migrating faster. Differential migration rate could indeed hint at differential adhesion and mechanical properties of the cells. 2. Hox genes have been shown to be upstream of and modulate adhesion molecules, which might be an interesting link. 3. So far, the authors have only analyzed expression of adhesion molecules at mRNA levels. However, the functional components are the adhesion proteins themselves. It might therefore be useful to stain embryos for some "obvious" candidate adhesion molecules, such as cadherins.If no further experiments are performed, then this should at least be discussed.

We thank the reviewer for raising this point and agree that while adhesion genes are not differentially expressed at the transcript level between HH4 and HH8 cells, the cells may indeed have different adhesion profiles/ characters and/or different mechanical properties. We believe this to be an interesting hypothesis for the differential behaviour of HH4 and HH8 tissue, with cell sorting via differential adhesion contributing to graft outcome. We have added a section on the possible role for cell sorting to the Discussion section (lines 360-392*)*.

- The authors describe a new, intermediate stage, namely cell dispersion, in which HH8 MSP pause when grafted into HH4 embryos. They perform scRNAseq and GO term analyses to analyze these cells in more detail. The also perform gene set enrichment analysis. However, I am still wondering about the exact identity of these cells. What are they? What markers do they express? Do they upregulate certain signalling pathways? Etc. I would for instance be interested if there are differences in FGF or Wnt levels/ activities. It would be useful if the authors could analyze their scRNAseq data further in this regard.

The original manuscript contains information on the gene expression of these cells in Figure 2C, in Supplementary Table 1 (now Supplementary File 1a) (which shows the 10 most DE genes in each cluster), and in Supplementary Figure 2 (now Figure 2 Supplement 1) (which shows feature plots for key marker genes). Additionally, the GO terms enriched amongst the full list of DE genes in cluster 6 can be found in Supplementary Table 2 (now Supplementary File 1b). We went back to these gene lists and do not note a signature relating to a particular signalling state; however several markers of primitive streak and early mesoderm were evident among the top differentially expressed genes. Since they do not express many of the clear markers of these states that can be seen in the annotated clusters, it suggests that they are in a transitory state that is also reflected in the position of grafted cell populations. We have added an additional description of the gene changes in the main text (lines 166-169).

- At several points in the manuscript, expression levels and patterns of HH4 and HH8 grafts are compared to each other.It does not become clear what the differences and similarities to the non-grafted cells of the same clusters are. Does grafting itself change the expression patterns?

In response to a similar question posed by reviewer 2, we have included an additional Supplementary Figure to support the transcriptomic equivalency of donor and host cells in the case of homochronic grafts. We initially direct the reviewer’s attention toward Figure 2G*,* where *eGFP*+ (donor) and *eGFP*- (host) cells are intermixed after clustering. This is also true of HH4-HH4 grafts at 3 hours after grafting (Figure 2 Supplement 2). These results suggest that grafting does not have any overt effect on the transcriptome of cells – or if it does, the effect is similar in host and donor cells.

- The authors found differences in cell cycle stages of HH4 and HH8 grafts. A more detailed discussion of this aspect would be useful rather than just excluding any cell cycle-related genes from the comparisons. Why could there be this difference? What effect could this have? Etc.

We appreciate this point. The differences between cell cycle representation in the HH4 and HH8 cells of clusters 6, 8 and 12 are subtle (Figure 3 Supplement 1 panels *D-E*). The choice to regress cell cycle associated gene expression was to prevent these genes eclipsing other genes of interest. We have added a few sentences to the discussion of the paper to address the possible significance of cell cycle differences (please see lines 351-352).

Optional: Other experiments that could increase the relevance of the work:- As discussed by the authors, they specifically compare HH4 stage to HH8, which represents primitive streak stage and 4-somite stage, respectively. It would therefore be interesting to perform grafts from HH8 to later stages, such as HH10, or vice versa, when the process of somitogenesis is more similar. This could reveal if their findings are specific for pre topost node development or more general. However, this might be outside of the scope of this study.

We agree that this is a very interesting question but also agree that it goes outside the scope of the study. We have mentioned a need for a greater diversity of grafting experiments (varied host and donor stages) in order to draw more general conclusions and principles of intrinsic and extrinsic cues in timing axis contribution. Indeed, it may well be the case that the relative importance of these cues varies across axis elongation, and/or that the relative “match” between host and donor tissue influences graft outcome. We have added text suggesting this as a future direction for the field within the discussion (line 388-392) and called for a greater range of grafts to determine whether the contributions of intrinsic and extrinsic cues to *Hox* expression vary across different timepoints in axis elongation (line 420-421).

Reviewer #3 (Significance (Required)):General assessment: This study provides a systematic analysis of the interaction of embryonic cell clusters from different developmental stages. To this end, "classical" developmental biology techniques, i.e. grafting (complicated techniques that probably less and less people can perform nowadays), is combined with more modern ways of analysis, i.e. scRNAseq. This allows the authors to dissect the differential behaviour of hetero- and homochonic grafts. In the longer term this data can provide the basis for further in-depth mechanistic analyses, some of which could be added here already. The involvement of Hox genes in the control of developmental time is interesting and should be placed into context of our current knowledge. Here, Hox genes are rather used as readout of developmental time rather than active players.Advance: How developmental time is maintained during embryonic development is a long-standing question in the field. This study provides conceptual advance in this question by describing a cell-intrinsic timer.Audience: This study is relevant for developmental biologists in general, since it describes how developmental time can be kept by a cell-intrinsic timer, at least in early stages of somite formation in chick embryos.My expertise: developmental biology, somitogenesis